# Phasic and tonic neuron ensemble codes for stimulus-environment conjunctions in the lateral entorhinal cortex

**Maryna Pilkiw[1], Nathan Insel[2,3], Younghua Cui[2], Caitlin Finney[2], Mark D Morrissey[2,4], Kaori Takehara-Nishiuchi[1,2,4]***

[1]Department of Cell and Systems Biology, University of Toronto, Toronto, Canada; [2]Department of Psychology, University of Toronto, Toronto, Canada; [3]Department of Psychology, University of Montana, Missoula, United States; [4]Neuroscience Program, University of Toronto, Toronto, Canada

**Abstract** The lateral entorhinal cortex (LEC) is thought to bind sensory events with the environment where they took place. To compare the relative influence of transient events and temporally stable environmental stimuli on the firing of LEC cells, we recorded neuron spiking patterns in the region during blocks of a trace eyeblink conditioning paradigm performed in two environments and with different conditioning stimuli. Firing rates of some neurons were phasically selective for conditioned stimuli in a way that depended on which room the rat was in; nearly all neurons were tonically selective for environments in a way that depended on which stimuli had been presented in those environments. As rats moved from one environment to another, tonic neuron ensemble activity exhibited prospective information about the conditioned stimulus associated with the environment. Thus, the LEC formed phasic and tonic codes for event-environment associations, thereby accurately differentiating multiple experiences with overlapping features.

**\*For correspondence:** takehara@psych.utoronto.ca

**Competing interests:** The authors declare that no competing interests exist.

## Introduction

The entorhinal cortex is thought to support rapid encoding of daily experiences by serving as an interface between the hippocampus and neocortical regions (*Eichenbaum, 2000*; *Squire, 1992*). In particular, lateral portions of the entorhinal cortex encode non-spatial information, such as objects (*Deshmukh et al., 2012*; *Deshmukh and Knierim, 2011*; *Keene et al., 2016*; *Tsao et al., 2013*) as well as visual and olfactory stimuli (*Igarashi et al., 2014*; *Leitner et al., 2016*; *Suzuki et al., 1997*; *Xu and Wilson, 2012*; *Young et al., 1997*). Recent evidence, however, demonstrates that the role of the LEC goes beyond a simple relay of non-spatial sensory information. For example, some cells in the LEC encode the location at which a specific object is placed (*Keene et al., 2016*) whereas other cells fire at the location of a previously encountered object or rewarding stimulus that is no longer present in the environment (*Deshmukh and Knierim, 2011*; *Tsao et al., 2013*). These findings are consistent with the LEC's role in recognizing objects in a specific context (*Hunsaker et al., 2013*; *Van Cauter et al., 2013*; *Wilson et al., 2013*) and suggest that the LEC may encode the content of events with the environmental context in which they take place.

The selectivity of the LEC for the combination of sensory event and spatial/environmental context raises a question as to how strongly environmental context *per se* modulates cell firing in the LEC and how it compares to the modulation by more transient sensory events. We set out to address these points by recording the activity of LEC cells during six blocks of trace eyeblink conditioning, in which rats associated either auditory or visual conditioned stimulus with eyelid shock in two different

**eLife digest** The context in which an event occurs plays a large role in how the brain understands and responds to the event. While a key part of context is where we are, contexts can also change within the same space: different meetings are held at different times and with different people in the same room, and a grassy field can be a place of intense competition or a place to relax and gaze at clouds. However, we have little understanding of how the brain sets up and maintains a sense of context.

A region of the brain called the lateral entorhinal cortex (LEC) responds to events as they happen, but may also maintain a record of past experiences, and helps us to learn new associations between events. To find out how LEC neurons might represent context, Pilkiw et al. measured the activity of individual LEC neurons in rats as they experienced different combinations of events and environments. In each trial, the rats were placed in one of two different rooms and exposed to one of two sensory cues (sound or light) six times, either alone or, to test learning, paired moments later with a mild stimulation to the eyelid. The gaps between the cues lasted from 20 to 40 seconds.

As expected, some LEC neurons responded to the sensory cues, and varied their responses to cues depending on whether or not they were paired with eyelid stimulation. What was much more striking is that almost all cells in the LEC behaved very differently in different contexts, both in response to the cues and also during the long gaps between the cues. This suggests that the LEC provides the brain with information about the circumstances of an event, and may be the reason we expect different events under different circumstances – even if we are in the same place.

We tend to underestimate how much we rely on context to remember events and to guide our behavior. Many disabling health conditions, including addiction, post-traumatic stress disorder and obsessive-compulsive disorder, are affected by context. For example, people who are trying to overcome drug addiction can often reduce their cravings by avoiding places and situations in which they previously used the drug in question. Understanding how the LEC represents context may therefore help us to develop treatments that target this brain region in order to alter harmful behaviors.

conditioning environments. This paradigm was chosen because the acquisition and retrieval of memory are known to depend on the LEC (*Morrissey et al., 2012*; *Ryou et al., 2001*; *Tanninen et al., 2013*, *2015*); thus, inferences could be made about the necessity of any observed activity patterns for behavior. The use of the transient, discrete sensory stimuli, rather than objects, allowed for precisely controlling the onset and offset of sensory events. We found that the LEC cells encode the conjunction of a sensory stimulus and an environment phasically during the stimulus presentations, while also differentiating between environments tonically based on the history of stimulus presentations in those environments, revealing that stimulus-environment conjunctions are coded in the LEC at multiple time scales.

## Results

### Behavioral performance

We recorded activity from cells in the LEC while seven rats underwent six blocks of trace eyeblink conditioning which required the subjects to form an association between a neutral conditioned stimulus (CS) and mild electric stimulation near the eyelid (US) over a 500 ms interval (*Figure 1A*). The conditioned responses (CRs) were monitored by recording electromyogram from the eyelid. To examine the selectivity of cell firing for stimulus modality and conditioning environment, three of these blocks (CS-US block) included the pairings of either auditory or visual CS and the US in one of two visually distinct conditioning boxes (*Figure 1B*). To examine the cells' selectivity for stimulus relationship, in the remaining three blocks (CS-alone block) the CS was presented by itself in the box. The CS presentations were separated with pseudorandom inter-trial intervals (ITI) ranging from 20 to 40 s. Each rat daily received six trial blocks in a fixed temporal sequence (an example pattern, *Figure 1C*; patterns used for each rat, *Table 1*), enabling the rats to acquire the temporal context

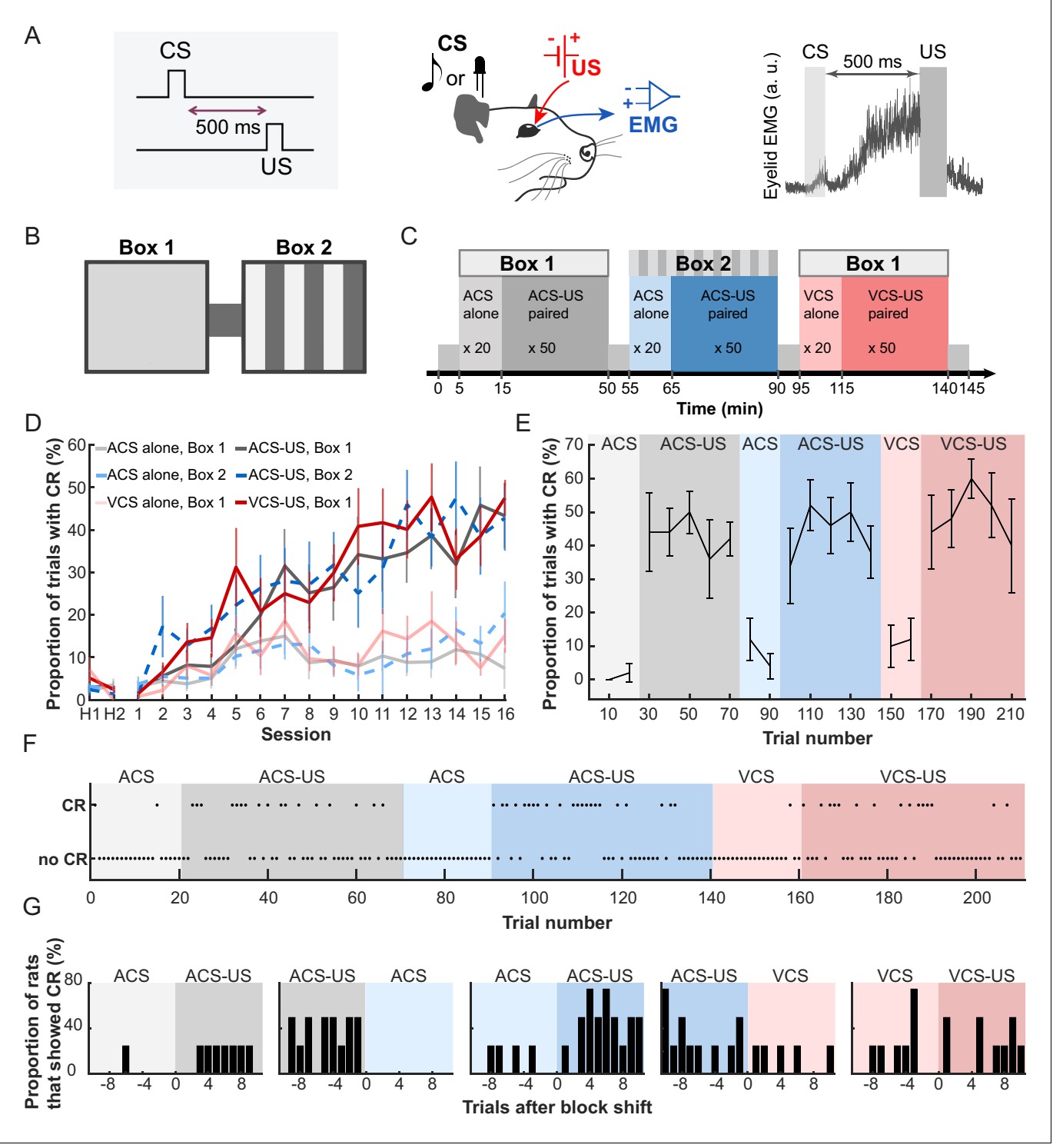

**Figure 1.** Rats acquired three associative memories with different stimulus and environmental features. (A) In trace eyeblink conditioning, animals associated a neutral conditioned stimulus (CS) with electrical shock near the eyelid (US) over a temporal gap (500 ms). With repeated pairings, rats developed anticipatory blinking responses before US onset (CRs), which were monitored by recording electromyogram (EMG) from the upper eyelid (a. u., arbitrary unit). (B) Daily conditioning took place in a conditioning chamber which consisted of two visually distinct rooms (Box 1 and 2) connected by a short walkway. (C) Rats first entered Box 1 and received an auditory stimulus (ACS) alone for 20 trials (CS-alone block), followed by 50 ACS-US pairings (CS-US block). They then moved to Box 2, where they received 20 ACS alone trials followed by 50 ACS-US paired trials. Finally, they returned to Box 1 and received a visual stimulus (VCS) alone for 20 trials followed by pairings of the VCS and US for 50 trials. (D) The frequency of CR expression increased in the CS-US paired, but not in the CS-alone, trials (mean CR% ± SEM; n = 7 rats; Sessions × Trial blocks interaction, $F_{75, 450} = 2.992$,

*Figure 1 continued on next page*

*Figure 1 continued*

p<0.001). H1 and H2 show the rate of blinking when only the CS was presented before any conditioning. (E) In the last session, CR% (in every 10 trials, mean ± SEM; n = 5 rats that received the CS-alone block before the CS-US paired blocks) showed an abrupt transition upon the shift from the block of CS-alone trials (light colors) to the block of CS-US paired trials (dark colors). (F) A trial-by-trial pattern of CR expression of Rat 2 during the last session. (G) The proportion of rats (n = 4 rats underwent the trial blocks in the same temporal sequence) that showed the CR in each trial during the last session.

The following figure supplement is available for figure 1:

**Figure supplement 1.** EMG activity during intervals between CS presentations.

predictive of what would happen in the present trial block. Consistent with our previous findings (*Morrissey et al., 2017*; *Takehara-Nishiuchi and McNaughton, 2008*), the rats gradually increased the expression of CRs in the CS-US paired trials but not CS-alone trials (*Figure 1D*; Two-way repeated measures ANOVA, Session $\times$ Block, $F_{75, 450} = 2.99$, p<0.001). The asymptotic level of CR expression during the three CS-US blocks was significantly different from that during the three CS-alone blocks (follow-up one-way repeated measures ANOVA on CR% during the last session, $F_{5, 30} = 21.9$, p<0.001, planned pairwise comparisons, ps < 0.05/6), but it was comparable between all three CS-US blocks (all ps > 0.7). In addition, the increased frequency of eyeblink responses was not observed during the ITIs of any of six trial blocks (*Figure 1—figure supplement 1A*; Session $\times$ Block interaction, $F_{70, 420} = 8.48$, p=0.717), suggesting that the eyeblink responses were conditioned to the CS, but not to the conditioning environment (see also, *Morrissey et al., 2017*). In the last session, the frequency of CR expression changed upon the transition from the CS-alone to the CS-US block within the first ten trials (*Figure 1E*), suggesting that blocks of CS-alone trials did not simply extinguish associations acquired on previous days, rather they formed a distinct temporal context between earlier and later trials. On a trial-by-trial basis, all except one rat responded correctly on the first trial of two blocks in which the change in stimulus contingency was signaled by the change in the conditioning environment (*Figure 1F*, the performance of Rat 2, *Figure 1G*, the performance of four rats that underwent the trial blocks in the same temporal order). In contrast, when the stimulus contingency was changed in the same environment, the rats gradually adjusted the frequency of CR expression over ten CS-US paired trials, suggesting that the presence of the US served as a cue for the block transition. Thus, the rats formed three associative memories that shared a common, relational feature (the CS-US association) but differed in a discrete, physical feature (the sensory modality of the CS) or environmental context (Box 1 or 2). They also learned to use the change in conditioning environment and the presence of the US to rapidly infer whether the CS would be paired with the US in the current block.

## Individual cells encoded sensory stimuli with environments where they were presented

From the first day of conditioning, action potentials of cells in the LEC were extracellularly recorded with a chronically-implanted microdrive array containing twelve independently movable, four-channel electrodes (tetrodes) and two reference electrodes. The final locations of the tetrodes were evenly

**Table 1.** Temporal order of the six trial blocks used for each rat.

| ID | 1ST | 2ND | 3RD | 4TH | 5TH | 6TH |
|---|---|---|---|---|---|---|
| RAT1 | ACS, Box 1 | ACS-US, Box 1 | ACS, Box 2 | ACS-US, Box 2 | VCS, Box 1 | VCS-US, Box 1 |
| RAT2 | ACS, Box 1 | ACS-US, Box 1 | ACS, Box 2 | ACS-US, Box 2 | VCS, Box 1 | VCS-US, Box 1 |
| RAT3 | ACS, Box 1 | ACS-US, Box 1 | ACS, Box 2 | ACS-US, Box 2 | VCS, Box 1 | VCS-US, Box 1 |
| RAT4 | VCS, Box 1 | VCS-US, Box 1 | ACS, Box 2 | ACS-US, Box 2 | ACS, Box 1 | ACS-US, Box 1 |
| RAT5 | ACS-US, Box 1 | ACS, Box 1 | ACS-US, Box 2 | ACS in Box 2 | VCS-US, Box 1 | VCS, Box 1 |
| RAT6 | ACS-US, Box 1 | ACS, Box 1 | ACS-US, Box 2 | ACS in Box 2 | VCS-US, Box 1 | VCS, Box 1 |
| RAT7 | ACS, Box 1 | ACS-US, Box 1 | ACS, Box 2 | ACS-US, Box 2 | VCS, Box 1 | VCS-US, Box 1 |

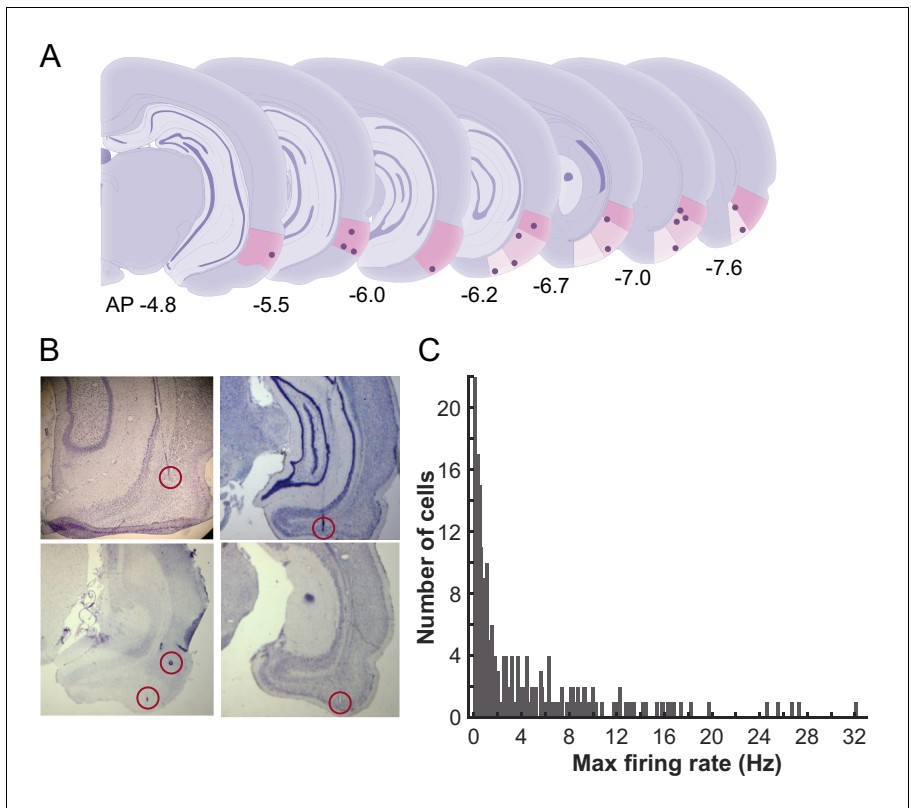

**Figure 2.** Recording locations and average firing rates of LEC units. (**A**) Schematic representations of the recording sites in the LEC along the anterior-posterior axis. The dots indicate final locations of tetrodes. The numbers show the distance from bregma (mm) (**B**) Example photomicrographs of the Nissl stained coronal sections with the tracks and final locations of the tetrodes (circled). (**C**) Histogram of the averaged firing rate (FR) of the recorded LEC cells. In each cell, FRs during inter-trial intervals were separately calculated in six trial blocks, and the highest FR was used as the FR for a cell.

distributed along the anterior-posterior axis within the LEC (*Figure 2A,B*). Firing rates of the majority of cells were less than 2 Hz (*Figure 2C*), which is consistent with those reported in the previous studies (*Deshmukh et al., 2012*; *Deshmukh and Knierim, 2011*; *Tsao et al., 2013*).

Among 250 cells recorded across seven rats (*Table 2*), 113 cells (45.2%) significantly changed their firing rate during the CS and subsequent intervals between CS offset and US onset relative to inter-trial intervals in at least one of six trial blocks (a random permutation test, $\alpha = 0.05$; *Table 3*). Of these CS-responding cells, 8.8% showed selectivity for CS modality, meaning that firing rates differed between the auditory CS and visual CS trials in Box 1, regardless of whether it was a CS-alone

**Table 2.** Number of recorded units and sessions in each rat.

| ID | Number of acquisition sessions | Number of sessions with recorded units | Total number of units |
| --- | --- | --- | --- |
| RAT 1 | 17 | 10 | 38 |
| RAT 2 | 16 | 12 | 61 |
| RAT 3 | 18 | 5 | 11 |
| RAT 4 | 16 | 13 | 49 |
| RAT 5 | 21 | 6 | 7 |
| RAT 6 | 20 | 13 | 55 |
| RAT 7 | 18 | 11 | 29 |

**Table 3.** Number and percentage of cells with selective CS-evoked firing for three variables. Table summarizes the number of cells that significantly changed CS-evoked firing rates depending on CS-US relationship (CS-alone trials vs. CS-US paired trials, R), CS modality (auditory CS vs. visual CS, M), conditioning environment (Box 1 vs. Box 2, E), the combination of them, or none of them (Non-S). The values in parentheses show the percentage of cells in each category to total CS-responding cells.

| Site | Non-S | R | M | E | R+M | R+E | M+E | R+M+E |
|---|---|---|---|---|---|---|---|---|
| Overall 113 | 28 (24.8%) | 11 (9.7%) | 10 (8.8%) | 7 (6.2%) | 13 (11.5%) | 8 (7.1%) | 14 (12.4%) | 22 (19.5%) |
| Superficial layers 56 | 16 (28.6%) | 5 (8.9%) | 5 (8.9%) | 3 (5.4%) | 7 (12.5%) | 4 (7.1%) | 7 (12.5%) | 9 (16.1%) |
| Deep layers 57 | 12 (21.1%) | 6 (10.5%) | 5 (8.8%) | 4 (7.0%) | 6 (10.5%) | 4 (7.0%) | 7 (12.3%) | 13 (22.8%) |

or CS-US paired trial (a random permutation test, α = 0.05; *Figure 3A*, Cell 1). In parallel, a separate set of CS-responding cells, 6.2%, was selective for the conditioning environment, exhibiting significantly different firing rates between the auditory CS presented in Box 1 and that in Box 2, regardless of whether it was presented with the US or alone (Cell 2). More cells (12.4%) differentially responded to the CS depending on both CS modality and conditioning environment (Cell 3). In parallel, 9.7% differentiated the firing response to the CS depending on whether it was presented alone or paired with the US at least in one of six trial blocks (Relationship, Cell 4). A considerable proportion of cells exhibited the selectivity for stimulus relationship together with the selectivity for stimulus modality (11.5%, Cell 5), conditioning environment (7.1%, Cell 6), or both (19.5%, Cell 7). Overall, the majority of CS-responding cells were selective for more than one task variable (*Figure 3B*). The proportion of cells in each selectivity category was comparable between superficial and deep layers of the LEC (*Table 3*). The magnitude of firing differentiation as measured by the distribution of a shuffle-corrected 'Differential index' (below 0-no significant to 1-strongest differentiation), was similar between stimulus modality and conditioning environment (*Figure 3C*, Kolmogorov-Smirnov test, p=0.330) while the magnitude of firing differentiation for stimulus relationship appeared to be weaker than those for modality (p=0.103) and environment (p=0.016). These results suggest that a sizable proportion of LEC cells transiently signaled the stimulus-environment conjunction time-locked upon the CS presentation.

## Nearly all cells encoded environments with the history of sensory stimuli presented in those environments

Most cells (137, or 54.8%) did not significantly change firing rates upon the onset of the CS relative to inter-trial intervals (ITI). These cells maintained stable firing rates throughout the entire period of each trial block, but nearly all of these cells (97.9%) changed ITI firing rates depending on which trial block a rat was in (*Figure 4A*). The across-block difference in their firing rates was not simply due to the difference in perceivable features of the conditioning environment. Virtually no cells (0.7%) were purely selective for conditioning environment: the cells generally did not exhibit significantly different firing rates in Box 1 blocks compared with Box 2, independent of ACS-US pairings or ACS alone. Rather, a subset of non-responding cells, 9.2%, was selective for both environment and stimulus relationship, exhibiting differential firing rates for the ACS-US block compared with the ACS-alone block in only one of the conditioning boxes (Cell 8, relationship and environment). Furthermore, a separate set of cells (25.9%) changed their ITI firing rates within the same conditioning environment depending on which of two CS was presented and whether the CS was paired with the US in a present trial block (Cell 9, relationship and modality). The largest proportion of cells (38.9%) was classified as selective for the stimulus relationship, stimulus modality, and conditioning environment, resulting in unique firing rates for one of six trial blocks (Cell 10) or different firing rates for each of six trial blocks (Cell 11). Importantly, their spike waveforms were comparable between six trial blocks (bottom of each panel), suggesting that the substantial changes in ITI firing rates across the trial blocks were not due to instability of the recording (*e.g.*, electrode drift).

The across-block difference in ITI firing rates was also observed in the CS-responding cells, and the proportion of cells for each category was comparable to that in the non-responding cells (*Table 4*). After CS-responding and non-responding cells were combined, the majority of these cells was classified as selective for more than two task variables (*Figure 4B*). The proportion of cells in each selectivity category was comparable between superficial and deep layers of the LEC (*Table 4*). The magnitude of firing differentiation, as measured by the distribution of a shuffle-corrected 'Differential index' (below 0-no significant to 1-strongest differentiation), was similar between stimulus modality and conditioning environment (*Figure 4C*, Kolmogorov-Smirnov test, p=0.787) while the magnitude of firing differentiation for stimulus relationship was weaker than those for stimulus

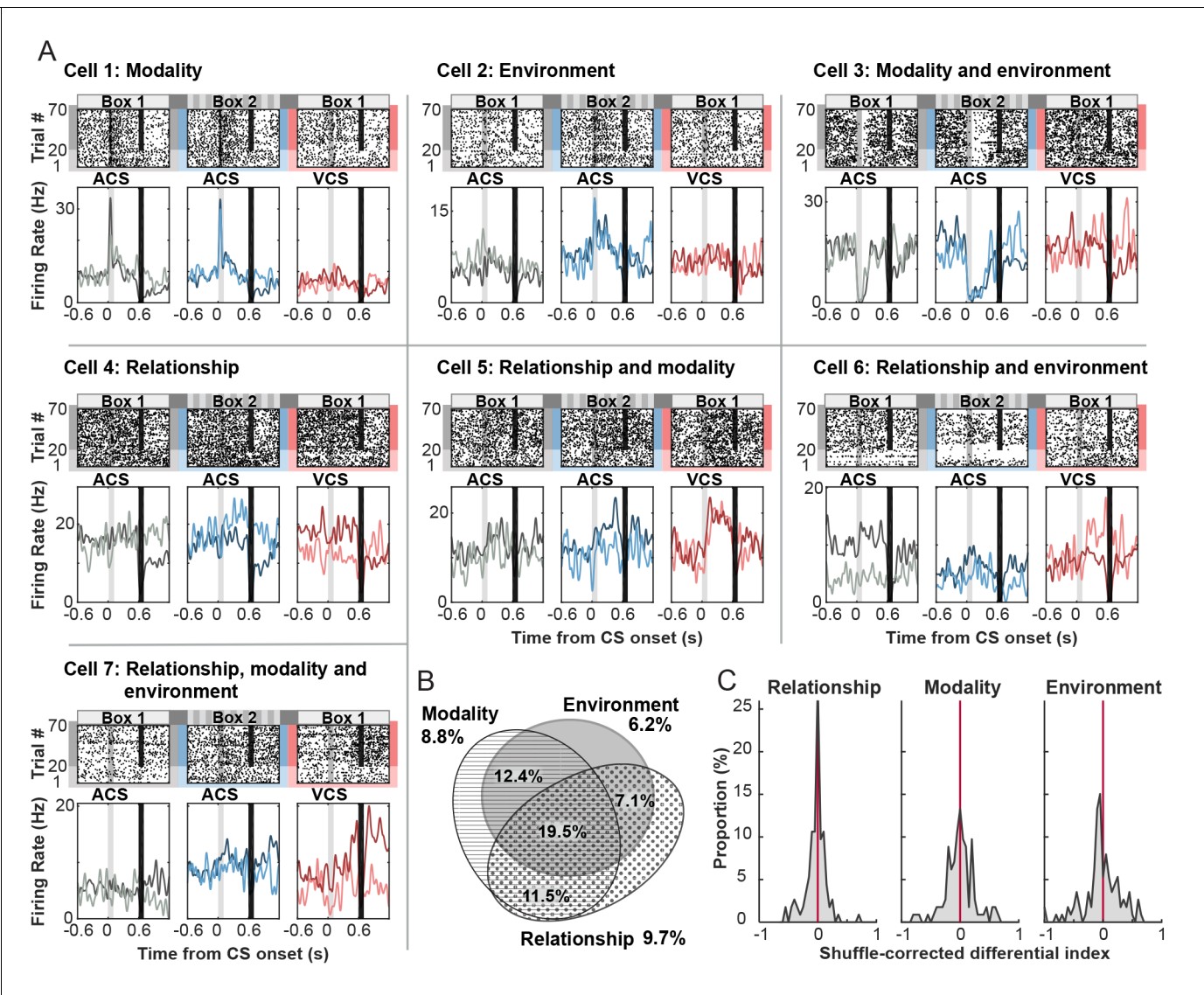

**Figure 3.** Phasic firing patterns selective for stimulus-environment conjunction. (**A**) Examples of firing patterns of cells that were identified as selective for stimulus relationship (CS alone vs. CS-US paired, Relationship), stimulus modality (auditory vs. visual CS, Modality), or conditioning environment (Box 1 vs. 2, Environment) and their combinations. Rasters and peristimulus time histograms (PSTH) represent activity patterns during the presentations of one of two CS (auditory, ACS or visual, VCS) in one of two conditioning environments (Box 1 or 2). Time 0 indicates CS onset. In each PSTH, the lines in light color show the firing pattern to the CS presented alone (the first 20 trials) while those in dark color show the pattern to the CS paired with the US (the latter 50 trials). Light gray shadings indicate the CS, and black bars mask the artifact induced by the US. (**B**) Area-proportional three-venn diagram (*Micallef and Rodgers, 2014*) showing a major overlap in selective cells for three task variables. The numbers show the percentage of cells in each category to total CS-responding cells. (**C**) Distributions of shuffle-corrected differential indices. A positive value means the significant selectivity (random permutation tests, α = 0.05).

modality (p=0.030) or conditioning environment (p=0.002). These findings suggest that regardless of firing rate changes during events, nearly all cells in the LEC signaled highly integrated information about sensory events taking place in a given environment during intervals between the events.

### Individual cells maintained selectivity for the combination of task features throughout the ITI period

To examine the stability of selective firing patterns during the ITI period, firing fluctuation within a trial block was compared with that across six trial blocks. Firing rates of each cell were stable during the ITI period in a trial block, but drastically differed between the trial blocks (*Figure 5A* left). The majority of cells (96.4%) showed a smaller degree of firing fluctuation, measured by Kullback-Leibler divergence, within each trial block than across the trial blocks (*Figure 5A* right, B). As a population, the degree of within-block fluctuation was significantly smaller than that of the between-block fluctuation (*Figure 5C*, signed rank test, p<0.001).

Another measure of firing stability was to examine the selectivity separately in a series of time bins during the ITI period. To test this, a methodological decision had to be made as to what temporal bin size was most appropriate for parsing the ITI. By re-running the test across multiple bin sizes, we found that the proportion of selective cells decreased as the bin size became shorter (*Figure 5D*), most likely due to the reduced accuracy in estimating firing rates. Importantly, the proportion of selective cells and the type of selectivity was stable across the time bins regardless of their temporal proximity to the CS onset (*Figure 5E*). These results suggest that firing rates of individual cells were largely homogeneous during the ITI period in a trial block.

### Firing rates of individual cells were not correlated with CR expression on a trial by trial basis

To examine whether individual cell firing was correlated with CR expression on a trial by trial basis, firing rates were compared between trials in which the rats showed CRs and those in which they did not. Across three blocks of CS-US pairings, 5.7–7.9% of cells' firing rates during CS-US intervals were judged to be selective for CR expression (2 out of 35 cells in ACS-US in Box 1; 4 out of 61 cells in ACS-US in Box 2; 5 out of 63 cells in VCS-US in Box 1). Virtually no firing rates during the ITI period were selective for CR expression (2 out of 35 cells in ACS-US in Box 1; 1 out of 61 cells in ACS-US in Box 2; 1 out of 63 cells in VCS-US in Box 1). These results suggest that although the LEC is necessary for memory acquisition and expression of this associative learning paradigm (*Morrissey et al., 2012*; *Tanninen et al., 2015*), firing patterns of individual cells were not correlated with behavioral expression of the memory on a trial-by-trial basis.

### LEC ensembles formed phasic and tonic codes that signaled physical stimulus features more strongly than environmental context

We next examined the selectivity of ensemble firing during trials and inter-trial intervals for sensory events and conditioning environment. Consistent with the observations from the single unit analysis (*Figures 3* and *4*), neuron ensembles appeared to form a unique firing pattern for each of six trial

**Table 4.** Number and percentage of cells with selective firing for task variables during intervals between trials. Table summarizes the number of cells that showed selective firing rates for three task variables during intervals between trials, as shown in **Table 3**. The values in parentheses show the percentage of cells in each category to total non-responding, CS-responding, or all cells.

| Response type/Site | Non-S | R | M | E | R×M | R×E | M×E | R×M×E |
|---|---|---|---|---|---|---|---|---|
| Non-responding 137 | 3 (2.2%) | 11 (8.0%) | 5 (3.6%) | 1 (0.7%) | 30 (21.9%) | 20 (14.6%) | 12 (8.8%) | 55 (40.1%) |
| CS-responding 113 | 4 (3.5%) | 10 (8.8%) | 4 (3.5%) | 2 (1.8%) | 13 (11.5%) | 15 (13.3%) | 10 (8.8%) | 55 (48.7%) |
| Superficial layers 110 | 5 (4.5%) | 11 (10.0%) | 5 (4.5%) | 1 (0.9%) | 19 (17.3%) | 15 (13.6%) | 11 (10.0%) | 43 (39.1%) |
| Deep layers 140 | 2 (1.4%) | 10 (7.1%) | 4 (2.9%) | 2 (1.4%) | 24 (17.1%) | 20 (14.3%) | 11 (7.9%) | 67 (47.9%) |

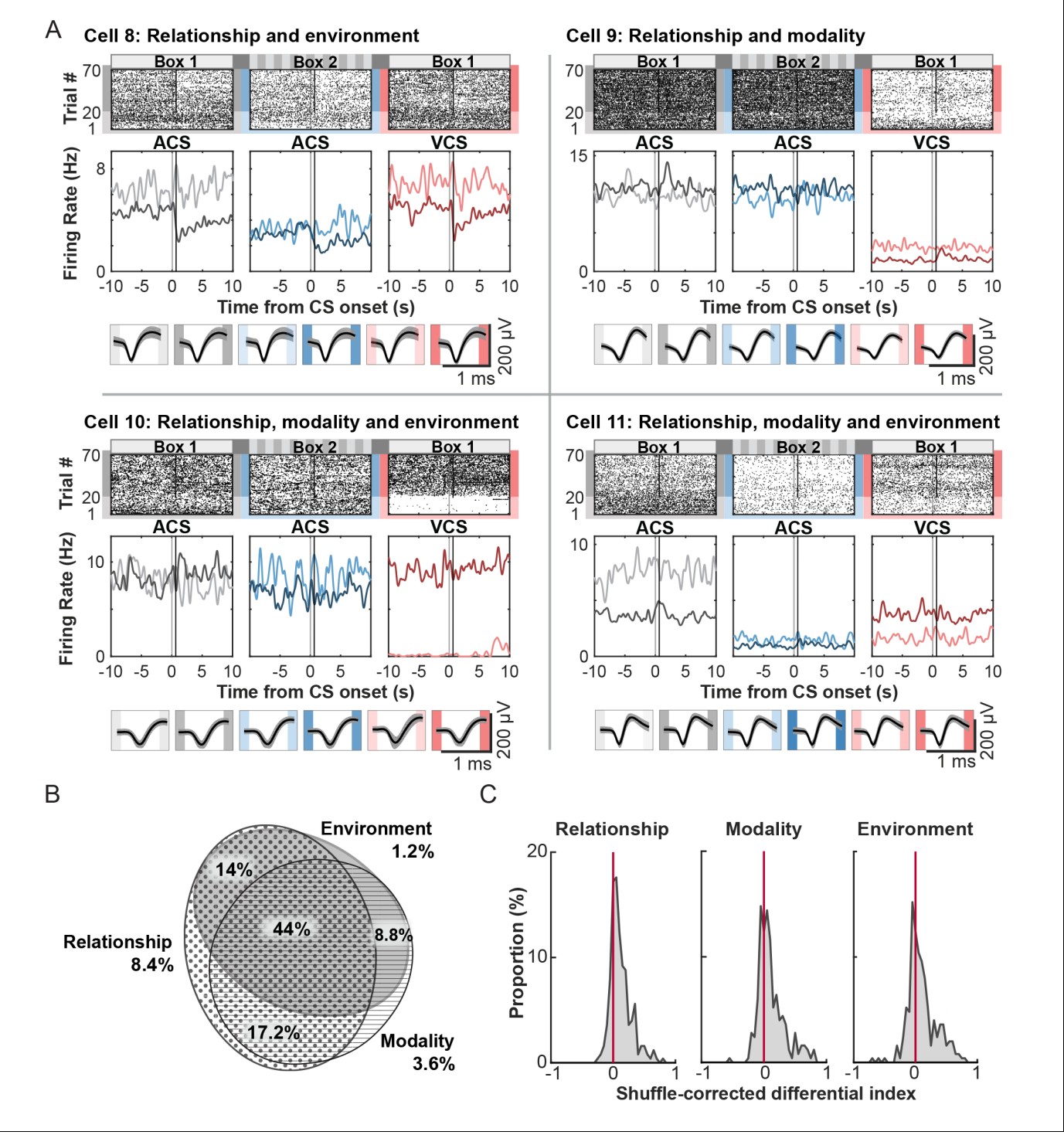

**Figure 4.** Tonic firing patterns selective for stimulus-environment conjunction. (A) Examples of firing patterns of cells during intervals between trials with rasters and peristimulus time histograms as in *Figure 3A*. The bottom panel shows each cell's average waveforms recorded in one of four wires of a tetrode during six trial blocks. (B) Area-proportional three-venn diagram showing that the majority of cells were selective for more than one task variable. The numbers show the percentage of cells in each category to all cells. (C) Distribution of the shuffle-corrected differential indices as in *Figure 3C*.

blocks, and the across-block difference was noticeable not only after, but also before the CS presentation (*Figure 6A*). To quantify the degree to which the ensemble activity differentiated each of

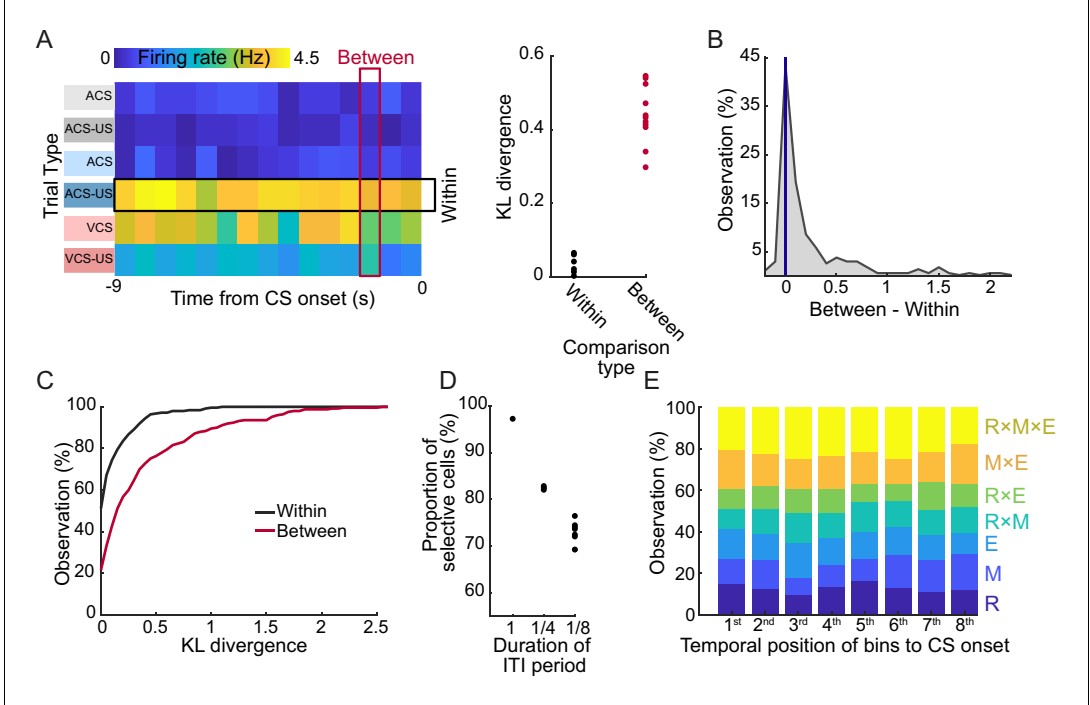

**Figure 5.** LEC cells stably maintained the selectivity for stimulus-environment conjunction during the stimulus-free period. (**A**) Example of fluctuations of binned firing rates (FR) of a cell across the 9 s interval during each of six trial blocks. The degree of FR fluctuation was marginal across time bins in each trial block ('Within', rows) while FR greatly changed across six blocks ('Between', columns) as evident in the difference in Kullback-Leibler divergence from a uniform distribution (KLD, right). (**B**) Distribution of the difference between the Between-KLD and Within-KLD. (**C**) Cumulative distribution of the Within- (black) and Between- (red) KLD. (**D**) Difference in the proportion of cells whose ITI firing rates were judged as selective for at least one of three task features depending on the bin size. The same criteria were applied to averaged firing rates in one 9 s bin (1), four 2.25 s bins (1/4), or eight 1.125 s bins (1/8) covering the 9 s ITI. (**E**) The proportion of selectivity categories (see **Table 4**) of firing rates during a series of eight 1.125 s bins during the 9 s ITI. The 8[th] bin was the closest to CS onset.

three task variables, the state vectors (i.e., the firing rates of 250 cells either during the trial or ITI period) were compared within versus between trial blocks (**Figure 6—source data 1**). During trials, ensemble firing patterns differentiated between ACS and VCS trials (Modality), between CS-alone trials and CS-US trials (Relationship), and ACS trials in Box 1 and 2 (Environment) more strongly than odd- and even-numbered trials of the same block (None, the upper limit; n = 20 randomly selected sets of 10 trials per trial block; one-way ANOVA, $F_{4, 195} = 238.975$, p<0.001; posthoc test Tukey HSD, all ps < 0.001). The degree of differentiation was the strongest for stimulus modality (vs. Environment, p<0.001; vs. Relationship, p<0.001) and was comparable to the differentiation of two trial blocks without any overlapping variables (All, the lower limit, p=0.122). Also, the differentiation of Relationship was weaker than the differentiation of Environment (p<0.001). During the ITI period, ensemble firing patterns differentiated two blocks with different stimulus modality more than those in different environments (one-way ANOVA, $F_{4, 195} = 142.092$, p<0.001; posthoc Tukey HSD, p<0.001), suggesting the stronger modulation by which CS had been presented than which conditioning box the rat was in. The differentiation for stimulus relationship was significantly weaker than that for stimulus modality (p<0.001) and conditioning environment (p<0.001) but was stronger than the chance-level differentiation (p<0.001).

Similar patterns were also observed when ensemble selectivity was quantified with a Support Vector Machine (SVM) classifier. During the trial and ITI periods, classification accuracy was well above chance level (accuracy above chance; trial, 87.1 ± 0.3%; ITI, 82.7 ± 0.5%, n = 20 runs with 200 randomly sampled cells), suggesting the strong differentiations of ensemble firing patterns across six trial blocks (**Figure 6—figure supplement 1A**). Subsequent binary discriminations of one of three task variables showed that in both the trial and ITI periods the classification accuracy for stimulus modality and conditioning environment was significantly higher than that for stimulus relationship

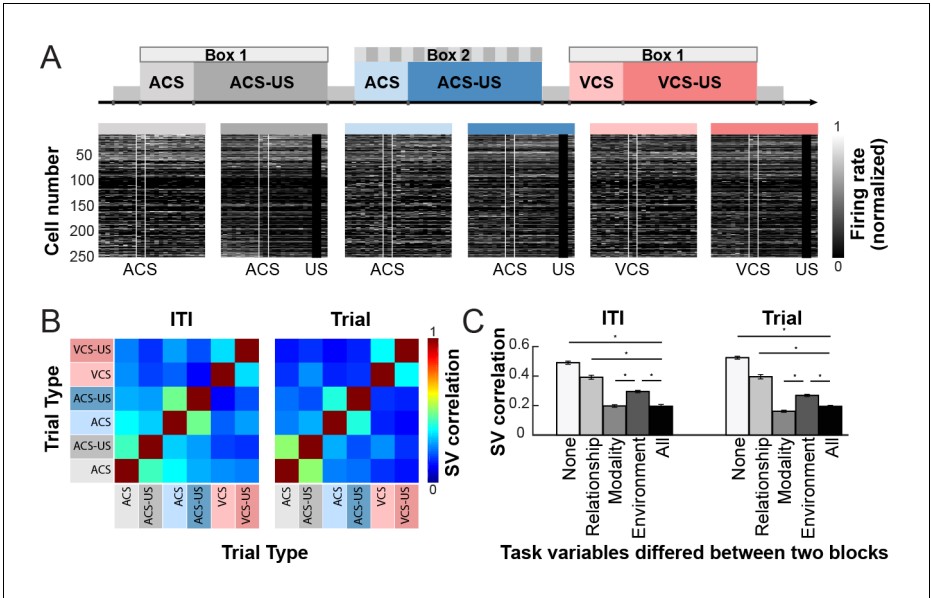

**Figure 6.** Ensemble activity showed a comparable level of selectivity for stimulus-environment conjunction during stimulus and non-stimulus periods. (**A**) Grayscale plots show the normalized firing rate during six trial blocks (from left, ACS trials in Box 1, ACS-US trials in Box 1, ACS trials in Box 2, ACS-US trials in Box 2, VCS trials in Box 1, and VCS-US trials in Box 1). Cells were sorted based on the ACS-induced firing rate during the ACS-US block in Box 1, from the largest increase (cell #1) to the largest decrease (cell #250). Two white lines indicate the onset and offset of the CS while black bars mask US artifacts. (**B**) Matrices of the correlation coefficient (r) of ensemble firing rates (State vector, SV) between two of six trial blocks during CS-US pairings (Trial) and intervals between trials (ITI). (**C**) During both task phases, the r for two blocks with different CS (Modality; mean ± SEM, n = 20 runs with 10 subsampled trials) was comparable to that for two blocks that differed in all task variables (All). It was significantly lower than that in different conditioning boxes (Environment) and with different stimulus contingencies (CS-alone blocks and CS-US blocks, Relationship). The r for Relationship was significantly higher than that for Environment but lower than that for odd- and even-numbered trials from the same block (None). *p<0.001, in posthoc Tukey HSD.

The following source data and figure supplement are available for figure 6:

**Source data 1.** Ensemble activity showed a comparable level of selectivity for stimulus-environment conjunction during stimulus and non-stimulus periods.

**Figure supplement 1.** Ensemble firing patterns differentiated trial blocks more strongly depending on the CS modality and conditioning environment than CS-US relationship.

(*Figure 6—figure supplement 1B*; n = 20 runs with 200 randomly sampled cells, t-test with Bonferroni corrections, trial, both ps < 0.001, ITI, both ps < 0.001). The classification accuracy for the stimulus modality was higher than that for the conditioning environment during the ITI (p=0.002), but not the trial period (p=0.693).

Collectively, the analyses with two independent measures of ensemble selectivity suggest that LEC formed ensemble codes that differentiated the six trial blocks not only during the trials but also during extended intervals between the trials.

## Tonic ensemble code prospectively signaled stimulus-environment conjunctions

Ensemble firing during ITIs maintained high selectivity for the modality of the CS and the CS-US relationship even though the CS had been terminated tens of seconds before. It is therefore possible that the LEC prospectively signals which CS will be subsequently presented, by recovering the stimulus-environment associations learned through past experiences. Alternatively, the LEC may retrospectively hold information about which CS had been presented several seconds ago. To

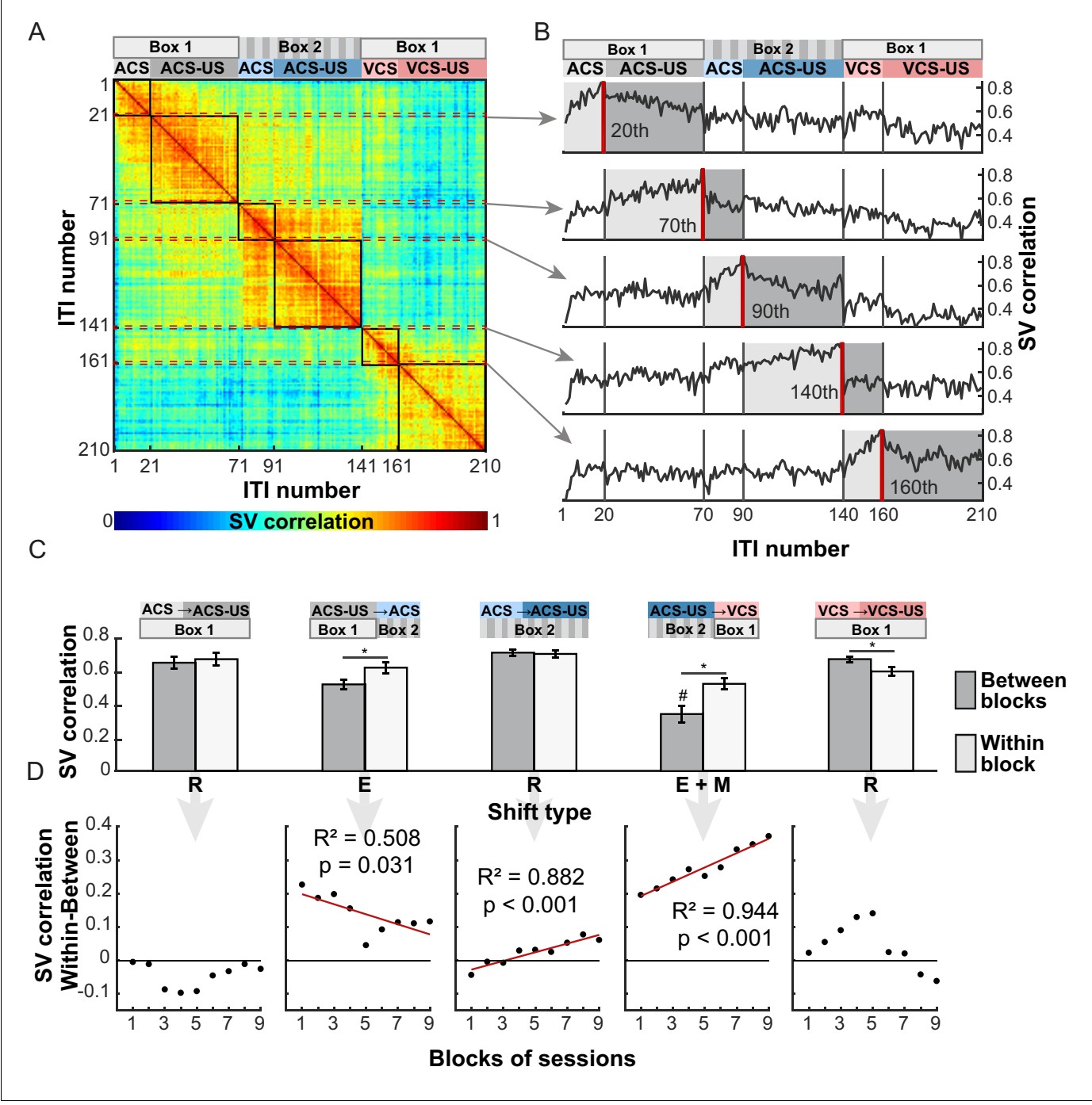

**Figure 7.** LEC ensemble activity showed an abrupt transition upon the shift of trial block involving changes in the conditioning environment and the stimulus modality. (A) A matrix of state vector (SV) correlation in which the element $r_{ij}$ represents the correlation coefficient between the SV computed during a 9 s interval (ITI) before the ith and jth CS presentation. The ITI number was continuously assigned to 210 ITIs according to the temporal order of six trial blocks as shown at the top of the matrix. The highlighted parts along the diagonal of the matrix (black squares) represent the r values between SVs within a trial block while the remaining parts represent those between different trial blocks. (B) The r for the last ITI in each trial block (red line) as a function of the temporal order of another ITI with which the r was computed. At the transition from the CS-alone block to the CS-US block in the same environment (20th, 90th, 160th), the r values with SVs in the same trial block ('Within-block', light-gray shading) was comparable to those with SVs in the following trial block ('Between-blocks', dark-gray shading). In contrast, the r value abruptly dropped upon the block shift including the change in the environment (70th) and CS modality (140th). (C) The Between-blocks r values (mean ± SD, n = 20 repeats with 100 subsampled cells) was significantly lower than the Within-block r values for two block shifts involving the environmental changes (E, E+M; *p<0.001 posthoc Tukey HSD), but not in the three shifts involving the stimulus contingency change (R). The Between-block r values were lower for the shift involving the changes in the CS

*Figure 7 continued on next page*

*Figure 7 continued*

modality and environment (E+M) than that involving the environmental change alone (E; #, p<0.001). (D) Over nine blocks of five sessions, the degree of ensemble transition (Within-block r minus Between-block r) increased for the block shift including the environmental and modality change (E+M), but decreased for the block shift including the environmental change alone (E).

The following figure supplement is available for figure 7:

**Figure supplement 1.** Changes in ensemble activity with experiences.

differentiate these possibilities, we examined the degree to which ITI ensemble firing changed upon the transition from one trial block to the next. This analysis was applied to 139 cells recorded from four rats which underwent six trial blocks in the same temporal sequence (*Table 1*). A state vector (firing rates of 139 cells) for an ITI (template) was compared against the vectors for the other ITIs using Pearson correlation coefficient (*Figure 7A*). Ensemble firing in an ITI was similar to those in the other ITIs within the same trial block, though the similarity appeared to monotonically decrease across ITIs within each trial block. In contrast to the gradual changes, ensemble firing showed an abrupt transition after the shift from the ACS-US block in Box 1 to the ACS-alone block in Box 2, (*Figure 7B*, template = the 70th trial) as well as the shift from the ACS-US block in Box 2 to the VCS alone block in Box 1 (template = the 140th trial). These block shifts involving the move from one box to the other resulted in a greater change in the ensemble firing (measured by a 'Similarity Score', 1-no change, 0-uncorrelated) than their spontaneous change within each block (*Figure 7C*, n = 20 runs with randomly sampled 100 neurons; t-test with Bonferroni corrections, ps <0.001). Notably, the block shift involving the change in CS modality and conditioning environment induced a greater change in ensemble firing than the block shift involving the environmental change alone (p<0.001). This difference was apparent during the last five recording sessions, but not during the first five recording sessions (*Figure 7—figure supplement 1*). Linear regression analyses showed that over nine blocks of five sessions, the degree of ensemble transition upon the block shift with the environmental and modality change was increased (*Figure 7D*, $R^2$ = 0.944, p<0.001) while the degree of transition upon the block shift with the environmental change alone was decreased ($R^2$ = 0.508, p=0.031).

In contrast with the abrupt transition of ensemble activity triggered by the environmental change, ensemble similarity gradually decreased over several ITIs upon the shift from the CS-alone to CS-US trial blocks within the same environment (*Figure 7B*, template = the 20th, 90th, and 160th trials). The change in ensemble firing during the first ITI was either smaller or comparable to the within-block change (*Figure 7C*, ACS block in Box 1, p=0.074; ACS block in Box 2, p=0.239; VCS block in Box 1, p<0.001). Also, in two out of three block transitions, the degree of ensemble change did not significantly increase or decrease with experiences (ACS block in Box 1, $R^2$ = 0.009, p=0.805; ACS block in Box 2, $R^2$ = 0.882, p<0.001; VCS block in Box 1, $R^2$ = 0.326, p=0.108). These results suggest that the transition of ensemble activity preceded the first presentation of the CS upon the block shifts involving the environmental change, but not the block shifts involving the contingency change within the same environment. And, in the former case, over two weeks of repeated exposures to the fixed temporal structure, the LEC ensembles gradually developed a greater sensitivity to the modality of the CS to be presented in the environment.

## Discussion

Accumulating evidence from behavioral and electrophysiological studies show that the LEC plays a critical role in associating events with the environments in which they take place (*Deshmukh and Knierim, 2011*; *Hunsaker et al., 2013*; *Keene et al., 2016*; *Tsao et al., 2013*; *Van Cauter et al., 2013*; *Wilson et al., 2013*). We show here that cells in the LEC signaled this event-environment conjunction not only during events ('phasic' coding) but also stably during extended periods between those events ('tonic' coding). The analyses of both single neurons and ensembles of neurons in the LEC revealed a highly selective code for conditioning epochs that were defined by both static, environmental features and associated transient stimuli. Selectivity was observed not only when stimuli were being presented but also, with comparable strength, during intervals between stimulus

presentations. These results suggest that the LEC may signal not only what is happening but also what had happened previously in an environment.

The identification of phasic and tonic codes was possible due to the types of stimuli that were used. Temporally precise, transient stimuli offered methodological advantages for discriminating between stimulus-evoked and stimulus-independent responses, thereby overcoming a limitation of many previous investigations which used physical objects to characterize neuron activity (*Deshmukh et al., 2012*; *Deshmukh and Knierim, 2011*; *Keene et al., 2016*; *Tsao et al., 2013*). In the present data set, ~45% of cells responded to the CS, which falls within the range of selective cells for unimodal sensory stimuli (47–90%, *Leitner et al., 2016*; *Suzuki et al., 1997*; *Xu and Wilson, 2012*; *Young et al., 1997*) but higher than that of cells selective for object location (~30%, *Deshmukh and Knierim, 2011*; *Tsao et al., 2013*). Some of these cells differentially responded to the CS depending on whether the CS was predictive of a US. Notably, these cells were also selective for the modality of the CS and environment in which the CS was presented (*Figure 3*). This conjunctive selectivity results in a clear separation of neural responses to the CS across six trial blocks (*Figure 6A*), suggesting that the cells carry both behaviorally relevant and incidental information about the CS. In contrast to the strong selectivity for the stimulus and environmental features of the task, almost no cells in the LEC differentiated firing patterns depending on CR expression on a trial by trial basis. This finding suggests that the LEC may not be involved in the generation of CRs per se. Rather the LEC may play a role in differentiating the CS depending on the current situation.

In parallel with stimulus-responsive cells, nearly all cells exhibited high selectivity for specific trial blocks by drastically discriminating between trial blocks during the non-stimulus periods (*Figure 4*). Ensembles were strongly selective for the trial blocks that used visual compared with auditory CS, yet only about half of these cells changed firing rates at the time of CS presentations and maintained stable firing patterns throughout the entire period of a trial block. Notably, virtually none of these cells were purely selective for the conditioning environment (independently from the stimulus features), suggesting that they did not simply respond to perceivable differences between two conditioning boxes, such as patterns of the wall and lighting (see also *Lu et al., 2013*). Moreover, given the absence of a significant difference in spontaneous eyeblink activity across six trial blocks (*Figure 1—figure supplement 1B*), differential firing rates were not reflective of variations of activity level across blocks. Therefore, a plausible interpretation would be that these cells encode the conditioning environment with the history of sensory events that took place in that environment.

The hypothesis that differential activity patterns between task epochs were driven by past events in those same contexts is consistent with prior observations of neural activity patterns in the LEC. In particular, prior experience with an object can later evoke selective firing patterns in the LEC even after that object has been removed: an 'object-trace' code (*Deshmukh and Knierim, 2011*; *Tsao et al., 2013*). In these earlier studies, object-trace selectivity was observed in ~15% of LEC cells (*Tsao et al., 2013*), while the present results showed that virtually all cells in the LEC differentiated sensory stimuli that had been experienced tens of seconds or longer in the past. This difference may be in part because our criteria for significance were based on the averaged firing rate across 9 s periods before CS presentations. This long time window serves as a kind of a 'low-pass-filter' that cancels out variations in spike timing across trials. In fact, the proportion of selective cells dropped down to ~50% when shorter bin sizes were used (*Figure 5D*). Another critical difference between our study and previous studies is the number of times stimuli were presented. Over about two weeks, the rats of the present study repeatedly underwent the same temporal structure of the trial blocks, each of which included seventy presentations of the CS. The total number of repeated stimulus presentations, therefore, is far greater than the repetition of object presentations used in the previous studies (*Deshmukh and Knierim, 2011*; *Tsao et al., 2013*). This task design could have facilitated the development of an internal model of the stimulus, which may be reflected in the high proportion of cells that maintain the selectivity during the stimulus-free periods.

The strong selectivity for the combination of sensory and environmental features unique to each trial block was also observed at the ensemble level during both the stimulus and stimulus-free periods (*Figures 6* and *7*). Notably, changes in ensemble activity during stimulus-free periods were tightly coupled with the shift of trial blocks with the change in the conditioning environment. When rats moved from one conditioning box to the other, ensemble activity showed abrupt transition *before* the first presentation of the CS (*Figure 7A–C*). Initially, the degree of ensemble transition was comparable regardless of whether the rats were to receive the different or same modality of the

CS from/as the one in the previous block. Over two weeks, however, it became greater for the block shift with the different CS than the one with the same CS (*Figure 7C,D*). This observation suggests that the LEC keeps track of what had happened in a given environment in the past and prospectively signals the information when the rat enters the environment next time. One may argue that LEC ensemble activity differentiated the trial blocks because of their temporal separation, rather than their difference in the stimulus and environmental features. In support of this view, some cells in the cortex change their responses to a sensory stimulus (*Fahy et al., 1993*; *Riches et al., 1991*; *Suzuki et al., 1997*) or environment (*Takehara-Nishiuchi et al., 2013*) over repetitive presentations of the stimulus, suggesting that they may keep track of the temporal order or repetition of experiences. In fact, within each trial block, the similarity of ensemble firing between stimulus-free periods decreased with time (*Figure 7B*). This time-dependent change in the ensemble activity, however, was substantially smaller than the changes upon the shift of two trial blocks in the different environment (*Figure 7B*), suggesting that time alone is not sufficient to account for the observed across-block changes in ensemble firing. Comparing the relative impact of time, environment, and past experiences on the LEC activity requires future studies which monitor the change of LEC ensemble activity while controlling the temporal order of experiences with and without changes in the environment or sensory events.

To disentangle the coding properties of the LEC that resulted in trial block selectivity, our correlation analysis compared a relative difference in the selectivity for three features (*Figure 6*). In general, ensemble selectivity was higher for stimulus modality (auditory vs. visual CS) compared with conditioning environment (Box 1 vs. 2). This observation could be considered alongside previous observations that LEC ensembles show greater selectivity for object identity compared with object location (*Keene et al., 2016*) and suggests that the physical features of sensory events have a strong influence on the neuron activity in the LEC. In parallel the selectivity of LEC ensembles was weaker for stimulus relationship (CS-US paired vs. CS alone) than stimulus modality. In our paradigm, stimulus relationship was a critical factor for adaptive behavior while stimulus modality and conditioning environment were not. The stronger selectivity for behaviorally irrelevant over relevant features sharply contrasts with the opposite selectivity observed in neuron ensembles in the medial prefrontal cortex (mPFC) during the same behavioral paradigm (*Morrissey et al., 2017*). These findings indicate a difference in information organization between the two regions, the LEC being more sensitive to incidental details of experiences and the mPFC being more sensitive to their behaviorally relevant features. The difference in feature selectivity may underlie the complementary roles that the hippocampal system and prefrontal cortex play for memory formation, consolidation, and assimilation (*Morrissey et al., 2017*; *Preston and Eichenbaum, 2013*).

Memories of daily experiences are thought to reside in a hippocampal-rhinal-neocortical network within which information becomes increasingly well integrated as it flows from the neocortical associative areas to the hippocampus (*Eichenbaum, 2000*; *Squire, 1992*). The presently observed selectivity of LEC cells suggests that the perceivable and relational information of experiences has already been highly integrated at the level of the LEC, even to a comparable level that was reported in the hippocampus (*Komorowski et al., 2013*; *Leutgeb et al., 2005*; *Moita et al., 2003*; *Wood et al., 1999*). The LEC, therefore, may not differ from the other regions regarding the level of information integration, but it may make a unique contribution to memory formation and retrieval by stably maintaining the highly integrated information throughout the experience. Such representations may serve as a scaffold in which efferent regions, such as the hippocampus, embed temporal dynamics of events within an experience, thereby facilitating accurately encoding the content of each experience. They also play a role in the retrieval of appropriate information based on the similarity between a present situation and situations encountered in the past.

In summary, the present findings support an emerging view that all regions within the hippocampal system encode the highly integrated information of sensory and environmental features of an experience (*Keene et al., 2016*). The LEC may be unique in that it signals the information not only during specific events but also stably throughout the experience. This coding property may support the LEC's involvement in the accurate binding of physical, relational, and contextual features of everyday experiences (*Eichenbaum, 2000*; *Morrissey and Takehara-Nishiuchi, 2014*; *Squire, 1992*).

## Materials and methods

### Animals

All surgical and experimental procedures were approved by the Animal Care and Use Committee at the University of Toronto. Seven male Long-Evans rats (Charles River Laboratories, St. Constant, QC, Canada) weighing 500–600 g at the time of surgery were used in the experiment. The rats were single-housed in Plexiglas cages in a colony room with *ad libitum* access to food and water. They were maintained on a reversed 12 hr light-dark circadian cycle, and all experiments took place during the dark part of the cycle. Prior to the experiment rats were handled 1–2 times per week.

### Electrode implants

Microdrives and electrodes were constructed in-house following a modified procedure described in a previous report (*Kloosterman et al., 2009*). A 3D-printed plastic base housed twelve individually movable tetrodes, two reference electrodes, and an electrode interface board (EIB-54-Kopf, Neuralynx, Bozeman, MT, USA). Tetrodes were made by folding in four and twisting a 36 cm wire (12 µm polyimide coated nichrome wire, Sandvik, Stockholm, Sweden). Each tetrode was connected to the plastic base with a shuttle containing a custom-made screw which allowed a precise control of the tetrode movement. From the top, individual tetrode wires were connected to the interface board with gold pins. From the bottom, the electrode tips were cut, and gold plated to reduce impedance to 200–250 kΩ. At the base, the tetrodes were arranged in a bundle with a honeycomb pattern, with the total area of the bundle not exceeding 1.2 × 1.2 mm.

During the surgery, the base of the drive with all tetrodes retracted was positioned at the surface of the brain in the right hemisphere (6.5 mm posterior, 5.3 mm lateral from Bregma at a 5° lateral angle) secured to the skull with a self-adhesive resin cement. During the surgery, tetrodes were lowered either 3 mm (rats 1–4) or 5–7 mm (rats 5–7). The reference electrodes were placed at the white matter tracts between the entorhinal cortex and the ventral hippocampus. Four stainless steel wires, two for delivering stimulation and two for recording electromyographic (EMG) activity, were implanted in the muscle of the left, upper eyelid and connected to the interface board. Two screws serving as a ground were placed in the skull above the cerebellum and the parietal cortex.

Across 7–10 days following the surgery, the rats were daily connected to the recording system to monitor the signal while tetrodes were gradually lowered to the lateral entorhinal cortex (LEC). The conditioning began at least seven days after the surgery. After each conditioning session, the tetrodes were lowered 30–65 µm to record different cells every day. Therefore, all cells were recorded only in one session. To allow for tetrodes to stabilize after each new placement and to avoid a potential electrode drift during the recording, the following recording session was conducted at least 20 hr after the electrode adjustment.

### Experimental apparatus and behavioral paradigm

Experimental apparatus consisted of two 20 × 20 × 70 cm boxes, connected with a 21.5 × 10 × 70 cm walkway (*Figure 1B*). The rats could freely move between the boxes and walkway when a partition between them was lifted by the experimenter. The first box was dark with uniformly painted walls. The second box was lit and had walls painted in a black and white striped pattern.

Daily recording sessions consisted of six blocks of classical trace eyeblink conditioning (*Figure 1C*) in which rats associated a neutral conditioned stimulus (CS) with an unconditioned stimulus (US) that was presented 500 ms after the offset of the CS. The CS was either 85 dB, 2.5 kHz pure tone (ACS, Multipurpose Sound Generator ENV-230, Med Associate Inc., St. Albans, VT, USA) or a white LED light pulsed at 50 Hz (VCS). The US was a mild electric stimulus applied near the left upper eyelid (a 100 Hz square pulse at 0.3–2.0 mA), generated by a stimulus isolator (ISO-Flex, A.M. P.I., Jerusalem, Israel). Seven rats received six blocks of this conditioning, each of which included one of two CS and took place in one of two visually distinct rooms (*Figure 1B*) in a fixed temporal sequence (*Figure 1C*, *Table 1*). For example, a rat was first placed in one of the rooms (Box 1) and presented with the ACS alone for 20 times, followed by 50 pairings of the ACS and US. The rat was then gently forced to walk over to the other room (Box 2) and received 20 presentations of ACS alone followed by 50 ACS-US pairings. Finally, the rat was forced to walk back to the original room and received 20 presentations of visual CS (VCS) alone followed by 50 pairings of VCS and US.

During the first two days, the rats underwent the same temporal sequence of the blocks without the presentation of the US.

## Data acquisition

Single-unit activity in the LEC and the electromyogram activity (EMG) in the eyelid were recorded by using a Cerebus neural signal processing system (Blackrock Microsystems, Salt Lake City, UT, USA). A rat was connected to the signal processing system through an electrode interface board (EIB-54-Kopf, Neuralynx, Bozeman, MT, USA) and the Omnetics headstage adapter (EIB-54K, Blackrock Microsystems). The signal was digitized at a headstage (CerePlex M 64, Blackrock Microsystems) at 30 kHz and transmitted to the signal processing system. The threshold voltage for signal acquisition of action potentials was set at 50–63 µV. And the signal exceeding the threshold in one of the four channels of a tetrode was recorded for 1 ms. The signal from tetrodes was amplified and filtered above 250 Hz. EMG activity was continuously sampled at 10 kHz and filtered between 250–5000 Hz.

## Behavioural analysis

Learning of the association between the CS and US was measured based on the frequency of adaptive conditioned responses (CR) which were defined as an increase in EMG amplitude prior to US onset (*Morrissey et al., 2012, 2017*). The instantaneous amplitude of the EMG signal was calculated as the absolute value of the Hilbert transform of the EMG signal. Then, two values were calculated in each trial: (1) the averaged amplitude of EMG signals during a 200 ms period before CS onset (pre-CS) and (2) the averaged amplitude during a 200 ms period before US onset or the corresponding interval in the CS-alone trials (CR value). A threshold was defined as the average of all pre-CS values plus one standard deviation. If the pre-CS value was greater than 130% of the Threshold in a trial, the trial was discarded due to the hyperactivity of a rat immediately before the CS presentation. If the pre-CS value was below the Threshold and CR-value was greater than 110% of the Threshold, the trial was considered to have a CR. Finally, in the case when the pre-CS value was between the Threshold and 130% of the Threshold, the trial was considered to have a CR only if the CR value above the Threshold was five times greater than the pre-CS value above the Threshold. CR% was calculated as a ratio of the number of trials containing the CR to the total number of valid trials (*i.e.*, total trials minus hyperactive trials). To examine whether eyeblink frequency that was not timelocked to the CS presentations increased with the conditioning, we applied the same analysis to EMG signals 480–280 ms before CS onset (pre-CR%). CR% and pre-CR% were calculated separately in each of six trial blocks. It was compared across sessions and conditions with a two-way repeated measures ANOVA with sessions and trial blocks as within-subjects factors.

To compare general activity level between the trial blocks, the amplitude of EMG trace was averaged across all inter-trial intervals of each trial block. Because the values were not distributed normally, they were converted to ranks and compared with a Kruskal-Wallis test.

## Data preprocessing

Putative units were isolated offline using an automated clustering software package (KlustaKwik, K. D., Harris, Rutgers, The State University of New Jersey, Newark, NJ, USA), followed by manual sorting (MClust, D.A. Redish, University of Minnesota, Minneapolis, MN; Waveform Cutter, S.L. Cowen, University of Arizona, Tucson, AZ, USA). Subsequent data analyses used only units in which the amplitude and shape of spike waveform were consistent across the entire recording period and the distribution of the inter-spike intervals did not contain more than 1% of the spikes within a 2 ms refractory period.

## Single-unit selectivity

Individual cells were first categorized into two types, CS-responding and non-responding cells. A cell was categorized as a CS-responding if its firing rate during the trial period (from CS onset to US onset, 600 ms) was significantly different from its firing rate during the period before CS onset (1 s). The significance was tested by random permutation tests which examined whether the observed firing rate difference fell within the 5% upper tail of the distribution of chance firing difference estimated by randomly assigning each firing rate to either the trial or the pre-CS period (1000 reassignments).

Subsequently, the selectivity of firing rates for three task variables was quantified as a Differential Index, which compared mean firing rates between two conditions:

Differential Index = (Fr1 – Fr2) / (Fr1 + Fr2)

where Fr1 and Fr2 are averaged firing rates across trials in two conditions. To test the selectivity of CS-evoked firing rates, firing rates during the 600 ms period from CS onset to US onset were used. To examine the selectivity of firing rates during intervals between trials, firing rates during the 9 s period before CS onset (ITI) were used. Because the duration of intervals between the trials ranged from 20 to 40 s, the ITI period started at least 11 s after US offset. This 9 s period was chosen to avoid any contamination of lasting firing rate changes that continued for a few seconds after US offset. For the selectivity for stimulus relationship, Fr1 was the mean firing rate during CS-alone trials, and Fr2 was the mean firing rate during the corresponding CS-US paired trials with the same CS and in the same conditioning environment. For the selectivity for stimulus modality, Fr1 was the mean firing rate during trials with the auditory CS in Box 1, and Fr2 was the mean firing rate during trials with the visual CS in Box 1. For the selectivity for the conditioning environment, Fr1 was the mean firing rate during trials with the auditory CS in Box 1, and Fr2 was the mean firing rate during trials with the auditory CS in Box 2. Raw differential indices were converted to absolute values. Then, in each cell the differential index at chance was estimated as 5% upper tail of its corresponding distribution of differential indices with shuffled trial labels (random permutation test, 1000 shuffles, $\alpha = 0.05$), and this value was subtracted from the value with the real trial label. A cell was judged as selective for a task variable if the shuffle-corrected differential index was greater than zero.

## Firing rate stability

During the 9 s ITI period, firing rates of each cell were binned into fifteen 600 ms bins and averaged across all trials in each of six trial blocks. The averaged, binned firing rates were divided by their sum, resulting in a distribution of firing rates across the bins. Kullback-Leibler divergence ($D_{KL}$) between this distribution (P) and a uniform distribution (Q) was calculated as follows:

$$D_{KL}(P||Q) = \sum_i P(i) \log_2 \frac{P(i)}{Q(i)}$$

As a measure for the within-block variation of firing rates, $D_{KL}$ was calculated across 15 bins in each of six trial blocks. As a measure for the across-block variation of firing rates, $D_{KL}$ was calculated across six trial blocks in each of 15 time bins. The median of the values was used as the within and across $D_{KL}$ for each cell.

## Correlation with CR expression

This analysis was applied to cells recorded during a block of CS-US trials in which a rat expressed the CR in minimum 20 trials (CR trials) and did not in minimum 20 trials (non-CR trials). To match the number of two types of trials, 20 trials were randomly sampled from each trial type. The Differential index was calculated with FR1 as firing rates during the CR trials and FRs as firing rates during the non-CR trials. The significance of the selectivity was determined by random permutation tests as shown in 'Single unit selectivity.'

## State vector analysis

To compare the selectivity of population firing patterns during trials for three task variables, we first constructed firing rate vectors that consisted of binned firing rates of all cells (state vectors) in each trial (CS + subsequent CS-US interval, 600 ms). In each block, ten state vectors were randomly selected and averaged, resulting in one state vector for each of six trial blocks. The firing rate of each cell in the state vectors was divided by its own maximum across the six blocks. We then calculated Pearson correlation coefficient (r) between the state vectors in two trial blocks with the ACS and VCS in the Box 1 (Modality, ACS alone vs. VCS alone, ACS-US vs. VCS-US), two blocks with the ACS in the Box 1 and 2 (Environment, ACS alone in Box 1 vs. ACS alone in Box 2, ACS-US in Box 1 vs. ACS-US in Box 2), or two blocks of ACS-alone trials and ACS-US paired trials in the Box 1 (Relationship, ACS alone vs. ACS-US, VCS alone vs. VCS-US). As controls, we calculated r values of state vectors between odd- and even-numbered trials within the same trial block (ACS-US paired in Box 1,

VCS-US paired in Box 1, an estimate of the highest r value) and pairs of trials without any overlapping task variable (ACS alone in Box 2 vs. VCS-US paired in Box 1, ACS-US paired in Box 2 vs. VCS alone in Box 1; an estimate of the lowest r value). We repeated these steps for twenty times, which generated 40 r values (2 pairs of trial blocks × 20 repeats) for each comparison. The same analysis was applied to firing rates during 600 ms bins randomly selected from 9 s periods before the CS onset (ITI period). The difference in the r values across these comparison types was quantified with a one-way ANOVA with posthoc comparison with Tukey HSD.

To examine how the ensemble activity changed upon the transition from one trial block to the next, we selected 139 cells recorded from four rats which underwent the trial blocks in the same temporal order (Rats 1, 2, 3, 7, see *Table 1*). We first randomly selected 100 cells out of 139 cells and constructed a state vector for each inter-trial interval (a 9 s period before CS onset). The firing rate of each cell in the state vectors was divided by its own maximum across all ITIs. Pearson correlation coefficient (r) was calculated for each state vector (template) with all other state vectors. This generated a 210 × 210 matrix of r values (*Figure 7A*). The change in the state vectors after the block shift was quantified with the averaged r values of the last trial in a trial block with the first 19 trials in the following block (Between-block Similarity Score). The change in the state vectors within a trial block was measured as the averaged r values of the last trial in a trial block and the 19 preceding trials (1st–19th CS-alone trials or the 31st to 49th CS-US trials) in the same block (Within-block Similarity Score). These steps were repeated for twenty times each of which used 100 randomly subsampled cells. The t-test with Bonferroni corrections was used to compare between the Between- and Within-block Similarity Scores in each comparison type or the Between-block Similarity Scores between the comparison types.

To examine how the degree of ensemble transition changed over about two weeks of recording sessions, 139 cells were sorted based on the session on which they were recorded. We then repeated the above-mentioned analysis on a series of 49 cells taken from the sorted 139 cells with an increment of 10 cells. This roughly corresponded to examining ensemble firing in a block of five sessions with an increment of one session. The Between-block Similarity Score was subtracted from the Within-block Similarity Score. Linear regression analyses were used to test whether the value changed over the session blocks.

## Population decoding analysis

As a secondary measure of ensemble selectivity, we used a machine learning algorithm, a Support Vector Machine (SVM) classifier (*Chang and Lin, 2011*; *Cortes and Vapnik, 1995*). The decoding accuracy of the classifier represented a degree to which ensemble firing patterns differentiated three task variables and their combination. The procedure was similar to that used in our previous study (*Morrissey et al., 2017*). SVM classification was performed in MATLAB (Mathworks, Natick, MA, USA) with the algorithms from the open source LIBSVM library (*Chang and Lin, 2011*). The SVM classifier constructs a model based on a set of attributes from the training data and then predicts the target values based on the test data attributes. In the present study, the attributes were normalized firing rates of a neuron population in a trial from one of the six trial blocks. The target values were the identities of the sampled trials.

Each run of SVM analysis used a set of 200 randomly subsampled cells (total 20 runs). Population firing rate vectors were constructed by concatenating the responses of these cells during a 600-ms period from CS onset to US onset (trial) or a 600-ms period randomly selected from a 9 s period before the CS onset (ITI) in a trial from one of six trial types. In each cell, the firing rate in each trial was divided by the maximum firing rate of the cell among all trials. Note that the cells were recorded in separate sessions from seven rats, and thus we ignored any correlated activity between cells. The classifier constructed a model with the Gaussian radial basis function kernels. To maximize the classification accuracy, two SVM parameters, cost and gamma, were identified by performing a grid search over a range of values, using the firing rates from all available trials for a given target value. Then, a new population firing rate matrix was generated from firing rates during twenty trials of each of six trial blocks. These trials were randomly drawn, without replacement, from all trials in a given trial block. Then, in each cell, the firing rate in each trial was divided by the maximum firing rate of the cell among the 120 trials (20 trials × 6 blocks). Half of the trials (10 trials from each block) were then used to train the SVM classifier with the parameters selected by the grid search. The remaining trials (10 trials from each block) were then used to test the decoding accuracy after

training. The process was repeated 20 times using a different sampling of 60 training and 60 test trials each time. The accuracy was defined as a proportion of correct predictions out of 1200 tests (60 test trials $\times$ 20 sets of randomly selected trials). To quantify the selectivity for one of three trial variables, the SVM classification was performed after collapsing six trial types into two trial types (for the modality feature, trials with the auditory CS and visual CS; for the environmental feature, trials in Box 1 and Box 2; for the relationship feature, CS-alone and CS-US paired trials). Classification accuracy was compared between three task features with t-test with Bonferroni corrections.

## Acknowledgements

This work was supported by NSERC Discovery Grant, CIHR Operating Grant, Ontario Research Fund Early Researcher Awards, CFI Leaders Opportunity Fund (KT), NSERC graduate fellowship (MP). The authors thank Stephanie Tanninen, Simone Cheng, and Patrick Gurges for their help in behavioral and histological experiments.

## Additional information

### Funding

| Funder | Grant reference number | Author |
| --- | --- | --- |
| Natural Sciences and Engineering Research Council of Canada | RGPIN-2015-05458 | Kaori Takehara-Nishiuchi |
| Canadian Institutes of Health Research | MOP-133693 | Kaori Takehara-Nishiuchi |
| Canada Foundation for Innovation | 25026 | Kaori Takehara-Nishiuchi |
| Natural Sciences and Engineering Research Council of Canada | 396157093 | Maryna Pilkiw |

The funders had no role in study design, data collection and interpretation, or the decision to submit the work for publication.

### Author contributions

MP, Software, Formal analysis, Funding acquisition, Investigation, Visualization, Writing—original draft; NI, Writing—review and editing; YC, CF, MDM, Investigation; KT-N, Conceptualization, Resources, Software, Formal analysis, Supervision, Funding acquisition, Investigation, Writing—original draft, Writing—review and editing

### Author ORCIDs

Maryna Pilkiw, http://orcid.org/0000-0003-1200-1708
Kaori Takehara-Nishiuchi, http://orcid.org/0000-0002-7282-7838

### Ethics

Animal experimentation: All surgical and experimental procedures were approved by the Animal Care and Use Committee at the University of Toronto (protocol number: 20011400).

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
