## [Decision Letter]

Thank you for submitting your article "Phasic and tonic neuron ensemble codes for stimulus-environment conjunctions in the lateral entorhinal cortex" for consideration by *eLife*. Your article has been favorably evaluated by Timothy Behrens (Senior Editor) and three reviewers, one of whom, Geoffrey Schoenbaum (Reviewer #1), is a member of our Board of Reviewing Editors. The following individual involved in review of your submission has agreed to reveal their identity: Thomas J. McHugh (Reviewer #3).

The reviewers have discussed the reviews with one another and the Reviewing Editor has drafted this decision to help you prepare a revised submission.

Summary:

The authors record in LEC during a trace eyeblink conditioning task in which an auditory and a visual cue led to shock in different contexts. LEC neurons exhibited both phasic and tonic firing, correlated with multiplexed information about the cues, their significance in different contexts and different trial blocks. Particularly remarkable, tonic activity reflected activity about the current block/context between trials, and this activity seemed to reflect a fairly precise internal model of the time or duration of the current block, inasmuch as it seemed to reflect the start of new blocks in the current context, which would otherwise be unsignaled.

Essential revisions:

Reviewers agreed that the work was sound and the results are quite novel and interesting. There were a couple things raised by the two reviewers and in discussions that seemed particularly worthwhile to do. One was to clarify whether firing changes really do precede the first trial of a new block in the same context or whether they are cued by the first shock or shock omission. The other was showing whether or not there is a relationship to behavior would be a good addition.

*Reviewer #1:*

In this paper, the authors record in LEC during a trace eyeblink conditioning task. In the task, the have an auditory and a visual cue, each of which predicts shock in a different context or box, and they alternate blocks of trials in which shock is given with blocks in which it is not. They report that LEC neurons exhibit both phasic and tonic firing, correlated with multiplexed information about the cues, their significance in different contexts and different trial blocks. Particularly remarkable, tonic activity reflected activity about the current block/context between trials, and this activity seemed to reflect a fairly precise internal model of the time or duration of the current block, inasmuch as it seemed to reflect the start of new blocks in the current context, which would otherwise be unsignaled.

I think the paper is generally sound and the results are quite novel and interesting. I have no major concerns from my reading and generally agree with the comments of the other reviewers. In particular, I think the authors should clarify whether firing changes really do precede the first trial of a new block in the same context or whether they are cued by the first shock or shock omission. I also think showing whether or not there is a relationship to behavior would be a good addition. Clarifying the other analysis issues would be important. In this area, I am particularly interested in the question raised about the decoding and why it is expressed as fractional performance rather than a raw percentage and the question of whether the tonic and phasic description is really a dichotomy.

*Reviewer #2:*

In this manuscript, Pilkiw and colleagues describe recordings made in the lateral entrorhinal cortex (LEC) as rats experienced a multi-block fear conditioning paradigm previously shown to depend of LEC function. The authors report that during cue presentation, neurons in LEC show complex patterns of mixed selectivity, encoding features of the task blocks (context, current meaning of the CS, modality of the CS) in a combinatorial way. Interestingly, the authors also demonstrate that information about the task's features was represented during the inter-trial interval by a separate population of neurons. The authors further show that the distinct representations of trial blocks they identify are not merely the result of a slow drift in ensemble representations over the course of a session, but rather that at least certain types of block transitions elicit step-like jumps in ensemble representations.

These data are an interesting and potentially valuable contribution to the literature. Attempts to understand entorhinal/hippocampal function in tasks that are not entirely and explicitly spatial are always welcome, and this paradigm is particularly interesting because of the rich temporal structure the blocked design creates; this allows the authors to investigate LEC encoding of both overtly-signaled aspects of the task (e.g. transitions between contexts) and more subtle features that require animals to track simultaneously multiple aspects of the task (e.g. whether the CS predicts shock, which depends on context, CS modality, and the temporal order of blocks). The analyses presented here are sophisticated, detailed, and generally sound, and the authors complement traditional single unit analyses with more modern, decoding-based approaches to identify what information is being represented by the population of neurons they recorded.

I have a few major questions that, if addressed, I think would strengthen the manuscript further.

1) The authors argue that animals learn the temporal structure of the task (i.e. whether the CS will be followed by the US in the particular block of trials). Is there any evidence of this specifically? When the block transition involves a change between the two contexts, rats clearly have overt evidence that there has been a change in block. However, is there specific evidence that they know before the first trial was completed whether or not they'd be shocked after the CS? Certainly once a shock does or does not arrive the identity of the block would be clear. But observing the context change as evidence of a new block and then waiting to see whether or not the CS in the current block is paired with shock is different than learning the temporal sequence of "shock present" and "shock absent" blocks.

This question is particularly interesting for block transitions that did not involve a change in context. In this case, if I understand the design correctly, there is no overt evidence whatsoever that a change in block has occurred. The first observable evidence that they're in a new block would be a change in whether the CS was or was not followed by the US. So, if rats did in fact learn the temporal structure of the task (and weren't using shock presence/absence to determine block identity), for block transitions that did not involve a change in context they would need to somehow time the length of blocks to estimate when transitions had occurred. Over the timescale of blocks on this task, that seems somewhat hard to imagine.

This also relates to the analyses in Figure 6 regarding transitions between neural representations of trial blocks. In the text of the manuscript the authors claim that ensemble transitions occur before the first CS presentation in a new block. Again, this is easy to understand in the case that animals moved to a new context for the block transition in question, but is it also true of transitions that did not involve a change in context? If pre-CS transitions in ensemble representations only occur when block switches involve a change in context, can it really be said that animals "prospectively" anticipated which block would be next?

To be clear, I'm not sure this issue substantially impacts the novelty or importance of the work here. Neural representations clearly differentiate task blocks and show step like changes across at least some block transitions; whether these representations are based on a fully-internal, learned model of how the task works, or whether they are prompted by the confirmatory presence/absence of shocks, I think the result is clear and interesting. But more clarity on precisely what is being claimed here would be appreciated.

2) The decoding results are a bit confusing to me. Why is classification accuracy presented as a fractional value of decoding performance on the shuffled data set rather than the raw percentage of trials correctly classified? I found this somewhat unintuitive. Does classification based on shuffled data not fall to the theoretical chance level? If not, does this suggest some bias has crept into the classifier somewhere?

Similarly, classification accuracy for "relationship" falls substantially below classification accuracy for shuffled data. How can performance on the actual data be worse than data where shuffling has removed all information about the actual trial identity?

3) The authors begin their analyses by dividing neurons into those that had a significant change in firing rate during the CS period and those that did not, and go on to conclude that the two classes of neurons form phasic and tonic codes related to the task. Is it clear, however, whether neurons with phasic responses did not also contribute to task representations during the ITI? Similarly, is it clear that the tonic neurons represented information homogeneously across the entire ITI? Perhaps some of the "tonic" responses were actually phasic and time-locked to the CS or US with a lag, such that their peak firing response occurred during the ITI. For instance, a hypothetical neuron might have reliably fired a burst of spikes 0.5 seconds after the offset of the US; while this response may have been perfectly phasic, it would have fallen during the ITI, and thus been counted as "tonic" activity. Because firing rates for the tonic neurons were computed over the full duration of the ITI (so far as I can tell), phasic responses like this hypothetical example could drive differences in the average ITI firing rate.

I grant that the example neurons the authors present do indeed look pretty "tonic", but it would be interesting to know whether the information the authors can decode during the ITI is truly homogenous across time at a single cell level, or is instead homogeneous across time at a population decoding level, but supported by transient, phasic activations of single neurons.

*Reviewer #3:*

The manuscript of Pilkiw et al. describes an experiment designed to characterize how neuronal responses in the lateral entorhinal cortex are modulated by sensory and contextual stimuli. The authors build on previous work from their group demonstrating activity in the LEC is required for the acquisition and expression of trace eyeblink conditioning, as well as their characterization of PFC neuronal activity during a very similar behavior. Here they design a clever protocol that interleaves multiple sensory stimuli and contexts to specifically examine how events and environments are associated in the LEC. Overall I find the experiment to be well designed and described and the analyses to support the informative and timely conclusions of the paper. I have a few suggestions and clarifications outlined below:

1) The authors suggest behavioral data establishes a framework in which the observed physiological activity can be interpreted as necessary for the behavior. While I do not disagree, it raises the issue of the correlation of the activity with the rat's behavior on a trial by trial basis. Given the rats only exhibit the CR on roughly 50% of the trials does a comparison of the activity of the various population (phasic/tonic; R/E/M?) of neurons during response/no response trials reveal the relative importance of one type of LEC coding in the rats behavioral output? I realize this is a bit off-topic for the main thrust of this study, it seems like an interesting analyses of these robust data.

2) The authors interpret the tonic activity related to context reflects the animals' experience in the task. Unlike in their PFC recording paper earlier this year, it appears in the current study recording made during both learning and post-learning periods were pooled. Was this indeed the case? If so, how does that impact the interpretation of the data in Figure 6 which is memory-driven predictive code?

3) Given previous LEC recording studies (Tsao et al., Deshmukh et al), albeit in a different task, it was a bit surprising to me that virtually every single recorded neuron here was responsive, either to the CS or tonically. The authors should address this discrepancy in the Discussion.

4) In the Materials and methods the authors describe the "Differential Index" as the absolute value of (FR1-FR2)/(FR1+FR2), thus running on a scale from 0 to 1. However in Figure 3 and Figure 4 they plot the values on a -1 to 1 scale. While I understand the value in this, it should be clarified in the Materials and methods.

5) A simple table describing the number of units recorded per rat and number of sessions per animal would be useful.

---

## [Author Response]

*Essential revisions:*

*Reviewers agreed that the work was sound and the results are quite novel and interesting. There were a couple things raised by the two reviewers and in discussions that seemed particularly worthwhile to do. One was to clarify whether firing changes really do precede the first trial of a new block in the same context or whether they are cued by the first shock or shock omission. The other was showing whether or not there is a relationship to behavior would be a good addition.*

*Reviewer #1:*

*[…] I have no major concerns from my reading and generally agree with the comments of the other reviewers. In particular, I think the authors should clarify whether firing changes really do precede the first trial of a new block in the same context or whether they are cued by the first shock or shock omission.*

We thank the reviewer for providing us with an opportunity to clarify this important aspect of our findings. Our result shows that upon the block shift that included the change in the conditioning environment, firing rate changes preceded the first presentation of the conditioned stimulus. Upon the block shift that included the change in stimulus contingency in the same environment, firing rates gradually changed over subsequent trials. To make these points clear, we edited the paragraph on this result in Results (subsection “Tonic ensemble code prospectively signaled stimulus-environment conjunctions”, last paragraph).

*I also think showing whether or not there is a relationship to behavior would be a good addition.*

We examined whether individual cells differentiated firing patterns between trials in which rats expressed conditioned responses (CR trials) and those in which they did not (non-CR trials). The proportion of cells that differentiated two trial types was at or below the cut-off for significance, suggesting that firing patterns of individual cells were not strongly correlated with CR expression on a trial by trial basis. To make this point clear, we added a new paragraph in Results (subsection “Firing rates of individual cells were not correlated with CR expression on a trial by trial basis”) and Materials and methods (subsection “Correlation with CR expression”).

*Clarifying the other analysis issues would be important. In this area, I am particularly interested in the question raised about the decoding and why it is expressed as fractional performance rather than a raw percentage and the question of whether the tonic and phasic description is really a dichotomy.*

In the original version of the manuscript, we reported classification accuracy above chance by subtracting the 95^th^ percentile of the chance distribution. If the value was greater than zero, classification accuracy with the real data was significantly better than chance. Because we realize that this value may not be intuitive, we revised the Results (subsection “LEC ensembles formed phasic and tonic codes that signaled physical stimulus features more strongly than environmental context”, second paragraph) and Figure 6—figure supplement 1.

To address the second point, we additionally performed and included a new analysis, which shows that both CS-responding and non-responding cells formed tonic selectivity for the event-environment conjunction. To make this point clear, we have rewritten the section on the tonic firing pattern (subsection “Nearly all cells encoded environments with the history of sensory stimuli presented in those environments”, last paragraph) and updated Table 4 and Figure 3. Throughout the manuscript, we removed any sentence that implies that a different set of cells shows the phasic and tonic selectivity.

*Reviewer #2:*

*[…] I have a few major questions that, if addressed, I think would strengthen the manuscript further.*

*1) The authors argue that animals learn the temporal structure of the task (i.e. whether the CS will be followed by the US in the particular block of trials). Is there any evidence of this specifically? When the block transition involves a change between the two contexts, rats clearly have overt evidence that there has been a change in block. However, is there specific evidence that they know before the first trial was completed whether or not they'd be shocked after the CS? Certainly once a shock does or does not arrive the identity of the block would be clear. But observing the context change as evidence of a new block and then waiting to see whether or not the CS in the current block is paired with shock is different than learning the temporal sequence of "shock present" and "shock absent" blocks.*

*This question is particularly interesting for block transitions that did not involve a change in context. In this case, if I understand the design correctly, there is no overt evidence whatsoever that a change in block has occurred. The first observable evidence that they're in a new block would be a change in whether the CS was or was not followed by the US. So, if rats did in fact learn the temporal structure of the task (and weren't using shock presence/absence to determine block identity), for block transitions that did not involve a change in context they would need to somehow time the length of blocks to estimate when transitions had occurred. Over the timescale of blocks on this task, that seems somewhat hard to imagine.*

We thank the reviewer for raising this important point. When the block shift occurred within the same environment, they adjusted the frequency of CR expression over the first ten trials (new Figure 1). This pattern suggests that the rats likely used the first presentation/omission of the US as a cue to infer the change in stimulus contingency. In contrast, all except one rat were able to adjust the response immediately upon the block transition involving the environmental change, suggesting that the rats were able to use the change in the conditioning environment to infer whether the CS would be paired with the US or not. To make these points clear, we added two new panels in Figure 1 and modified the corresponding section in Results (subsection “Behavioral performance”).

*This also relates to the analyses in Figure 6 regarding transitions between neural representations of trial blocks. In the text of the manuscript the authors claim that ensemble transitions occur before the first CS presentation in a new block. Again, this is easy to understand in the case that animals moved to a new context for the block transition in question, but is it also true of transitions that did not involve a change in context? If pre-CS transitions in ensemble representations only occur when block switches involve a change in context, can it really be said that animals "prospectively" anticipated which block would be next?*

*To be clear, I'm not sure this issue substantially impacts the novelty or importance of the work here. Neural representations clearly differentiate task blocks and show step like changes across at least some block transitions; whether these representations are based on a fully-internal, learned model of how the task works, or whether they are prompted by the confirmatory presence/absence of shocks, I think the result is clear and interesting. But more clarity on precisely what is being claimed here would be appreciated.*

Ensemble firing patterns abruptly changed before the first presentation of the CS only in the block shift involving the change in the conditioning environment. Importantly, the degree of ensemble transition was greater in the block shift involving the change of the conditioning environment and CS modality than the shift involving the environmental change alone. This result suggests that the LEC ensembles prospectively signal the modality of the CS associated with the present environment before the first CS presentation. In the block shift involving the stimulus contingency change within the same environment, ensemble activity patterns gradually changed over ~ten trials. To make these points clear, we edited the corresponding paragraph in Results (subsection “Tonic ensemble code prospectively signaled stimulus-environment conjunctions”, last paragraph).

*2) The decoding results are a bit confusing to me. Why is classification accuracy presented as a fractional value of decoding performance on the shuffled data set rather than the raw percentage of trials correctly classified? I found this somewhat unintuitive. Does classification based on shuffled data not fall to the theoretical chance level? If not, does this suggest some bias has crept into the classifier somewhere?*

The distribution of classification accuracy with shuffled data peaks closely at the theoretical chance level, and its width allows for determining whether classification accuracy with the real data is significantly better than chance. In our original analysis, we used the 5% upper tail of the chance distribution as the cut-off for significance and subtract it from classification accuracy with the real data. Because we realize that this way of data presentation may not be intuitive, we revised the Results and the figure so that they report raw classification accuracy (subsection “LEC ensembles formed phasic and tonic codes that signaled physical stimulus features 201 more strongly than environmental context”, second paragraph, Figure 6—figure supplement 1).

*Similarly, classification accuracy for "relationship" falls substantially below classification accuracy for shuffled data. How can performance on the actual data be worse than data where shuffling has removed all information about the actual trial identity?*

The threshold from the shuffled data is the 95th percentile line; thus, 95% of “chance” classifications will fall below this line. The significance of this when looking at the real data is, simply, that the classifier was unable to discriminate the relationship cues at a level that exceeded statistical significance. A “significantly worse than chance” discrimination would require the true data to fall below a different threshold, such as the 5^th^ percentile mark, which was not observed.

*3) The authors begin their analyses by dividing neurons into those that had a significant change in firing rate during the CS period and those that did not, and go on to conclude that the two classes of neurons form phasic and tonic codes related to the task. Is it clear, however, whether neurons with phasic responses did not also contribute to task representations during the ITI?*

We thank the reviewer for raising this important point. We applied the same analysis on stimulus-free, ITI firing rates of CS-responding cells and found that the majority of them (96.5%) were selective for at least one of three task features. Therefore, the tonic and phasic codes were not opposing one-another: the phasic code was supported by a subset of the population supporting the tonic code. In other words, nearly all cells changed ITI firing rates depending on the trial blocks while only a subset of them additionally changed their firing rate patterns upon the stimulus presentations. To make these points clear, we have rewritten the section on the tonic firing pattern so that it reports the selectivity of ITI firing rates of both CS-responding and non-responding cells (subsection “Nearly all cells encoded environments with the history of sensory stimuli presented in those environments”, last paragraph). We have also updated Table 3 and Figure 3 accordingly. Throughout the manuscript, we removed any sentence that implies that a different set of cells shows the phasic and tonic selectivity.

*Similarly, is it clear that the tonic neurons represented information homogeneously across the entire ITI? Perhaps some of the "tonic" responses were actually phasic and time-locked to the CS or US with a lag, such that their peak firing response occurred during the ITI. For instance, a hypothetical neuron might have reliably fired a burst of spikes 0.5 seconds after the offset of the US; while this response may have been perfectly phasic, it would have fallen during the ITI, and thus been counted as "tonic" activity. Because firing rates for the tonic neurons were computed over the full duration of the ITI (so far as I can tell), phasic responses like this hypothetical example could drive differences in the average ITI firing rate.*

*I grant that the example neurons the authors present do indeed look pretty "tonic", but it would be interesting to know whether the information the authors can decode during the ITI is truly homogenous across time at a single cell level, or is instead homogeneous across time at a population decoding level, but supported by transient, phasic activations of single neurons.*

To address the reviewer’s question, two new analyses have been added to the manuscript. First, we quantified the fluctuation of firing rates during the ITI period by calculating Kullback-Leibler divergence (KLD) between a uniform distribution and fifteen 600-msec binned firing rates during the ITI period in a trial block (within-KLD). As a comparison, we also calculated KLD between a uniform distribution and the binned firing rate across six trial blocks (between-KLD). We found that in the majority of cells (96.4%) the within-KLD was smaller than the between-KLD (new Figure 5), suggesting smaller firing fluctuations across time bins in each trial block than across six blocks in each time bin. Second, we divided the ITI period into smaller bins (1/4 or 1/8 of the 9-sec period) and examined the selectivity of firing rates separately in each of these time bins. The proportion of selective cells decreased as the bin size became smaller (new Figure 5), likely due to the reduced accuracy in firing rate estimation. Across the time bins, the proportion of each selectivity category was almost identical (new Figure 5). These results support our view that individual cells maintained the selective firing patterns stably throughout the ITI period. To make these points clear, we added a new paragraph in Results (subsection “Individual cells maintained selectivity for the combination of task features throughout the ITI period”) and Materials and methods (subsection “Firing rate stability”) along with one new figure (Figure 5).

As suggested by the reviewer, some cells indeed showed lasting firing rate changes after US offset for a few seconds. Because we defined the start of the ITI period minimum 10 seconds after US offset, these phasic, US-evoked firing rate changes did not affect our analysis. To make this point clear, we added several sentences in Materials and methods (subsection “Single-unit selectivity”, last paragraph).

*Reviewer #3:*

*[…] I have a few suggestions and clarifications outlined below:*

*1) The authors suggest behavioral data establishes a framework in which the observed physiological activity can be interpreted as necessary for the behavior. While I do not disagree, it raises the issue of the correlation of the activity with the rat's behavior on a trial by trial basis. Given the rats only exhibit the CR on roughly 50% of the trials does a comparison of the activity of the various population (phasic/tonic; R/E/M?) of neurons during response/no response trials reveal the relative importance of one type of LEC coding in the rats behavioral output? I realize this is a bit off-topic for the main thrust of this study, it seems like an interesting analyses of these robust data.*

To address the reviewer’s point, we compared firing rates of individual cells between trials in which rats expressed conditioned responses (CR trials) and those in which they did not (non-CR trials). This analysis required us to match the number of two types of trials (20 trials minimum per trial type). Because ~1/4 of sessions met this criterion, we were only able to use 35-63 cells (the number varied across three CS-US trial blocks) for this analysis. Among these limited samples, the proportion of cells that differentiated two trial types was at or below the cut-off for significance, suggesting that firing patterns of individual cells were not correlated with CR expression on a trial by trial basis. To make this point clear, we added a new paragraph in Results (subsection “Firing rates of individual cells were not correlated with CR expression on a trial by trial basis”), Discussion (second paragraph), and Materials and methods (subsection “Correlation with CR expression”).

*2) The authors interpret the tonic activity related to context reflects the animals' experience in the task. Unlike in their PFC recording paper earlier this year, it appears in the current study recording made during both learning and post-learning periods were pooled. Was this indeed the case? If so, how does that impact the interpretation of the data in Figure 6 which is memory-driven predictive code?*

We would like to thank the reviewer for inspiring closer examination into experience effects, as we believe this has strengthened the study. Our original manuscript did not analyze changes with learning phase for several reasons. First, in our previous study, learning and post-learning periods were determined by when CR% reached 60%. Because CR% of the rats in the current study only reached ~45% at the end of about two weeks of training, all sessions in the current study would fall under the “learning” stage category. Note that the lower performance in the current set of rats is likely because the ratio of CS-US trials relative to CS-alone trials was lower in the current design (5:2) compared with the previous design (4:1). Also, due to technical difficulty, we were able to record spike activity from the LEC only for ~16 days, which is about half of a total number of the recording sessions in our previous study.

To address the reviewer’s point, we performed an additional analysis that examined changes of ensemble activity over time or experience, rather than changes with learning per se. Sessions were divided into nine blocks of five sessions each, and the degree of ensemble transition was compared across the session blocks. Initially, the degree of ensemble transition was comparable between two types of the block shift involving the change in the conditioning environment. Over the course of two weeks, the degree of ensemble transition became greater upon the block shifts with the changes in both the environment and CS modality, but not the block shifts with the environmental change alone (new Figure 7). These results suggest that the LEC ensembles become capable of prospectively signaling the stimulus to be presented in a given environment only after repeated exposures to the fixed temporal sequence of trial blocks. We believe this result significantly strengthens our argument that the LEC develops the prospective code with experiences and thank the reviewer for raising this important point. The result of this analysis was added in Results (subsection “Tonic ensemble code prospectively signaled stimulus-environment conjunctions”) and Materials and methods (subsection “State Vector Analysis”, last paragraph), along with two new figures (Figure 7, and Figure 7—figure supplement 1).

*3) Given previous LEC recording studies (Tsao et al., Deshmukh et al), albeit in a different task, it was a bit surprising to me that virtually every single recorded neuron here was responsive, either to the CS or tonically. The authors should address this discrepancy in the Discussion.*

We thank the reviewer for raising this important point of discussion. The proportion of CS-responding cells (~45%) falls in the range of selective cells for unimodal sensory stimuli (47-90%, Leitner et al., 2016; Suzuki et al., 1997; Xu and Wilson, 2012; Young et al., 1997) but higher than that of cells selective for object location (~30%, Deshmukh and Knierim, 2011; Tsao et al., 2013). In parallel, nearly all cells in the LEC showed the selectivity for the CS that were presented tens of seconds ago, which was much higher than cells selective for locations at which an object was placed in the past (~15%; Tsao et al., 2013). This difference may be in part because our criteria for the significant selectivity were based on the averaged firing rate across 9-sec period within the ITI. This long period allows for accurately estimating firing rates by effectively canceling out fluctuations of spike timing across trials. Another reason may be that the total number of repeated CS presentations was far greater than the repetition of object presentations in the previous studies (Deshmukh and Knierim, 2011; Tsao et al., 2013). This task design would facilitate the development of an internal model of the CS, which may be reflected in the high proportion of cells that maintained the selectivity during the stimulus-free periods. To make these points clear, the new section has been added to Discussion (second and fourth paragraphs).

*4) In the Materials and methods the authors describe the "Differential Index" as the absolute value of (FR1-FR2)/(FR1+FR2), thus running on a scale from 0 to 1. However in Figure 3 and Figure 4 they plot the values on a -1 to 1 scale. While I understand the value in this, it should be clarified in the Materials and methods.*

Thank you for the careful review. To test the significance of Differential Index (DI), in each cell, we conducted a random permutation test based on the distribution of DI at chance. The 95^th^ percentile of the chance distribution was subtracted from raw DI, resulting in a “shuffle-corrected” DI that became positive if the DI was significantly different from the chance. To make this point clear, we edited the section in Materials and methods (subsection “Single-unit selectivity”, last paragraph) and renamed the x label of Figure 3 and Figure 4.

*5) A simple table describing the number of units recorded per rat and number of sessions per animal would be useful.*

According to the reviewer’s suggestion, we added a new table (Table 4).